

# Mid-field tsunami hazards in greater Karachi from seven hypothetical ruptures of the Makran subduction thrust

Haider Hasan[1], Hira Ashfaq Lodhi [2], Shoaib Ahmed [1,3], Shahrukh Khan[4], Adnan Rais[1], Muhammad Masood Rafi[5]

5 [1] Department of Civil Engineering, NED University of Engineering and Technology, Karachi, 75270, Pakistan
[2] Department of Physics, NED University of Engineering and Technology, Karachi, 75270, Pakistan
[3] Department of Agricultural and Biosystems Engineering, Iowa State University, Ames, IA 50011-3270, US
[4] Department of Urban and Infrastructure Engineering, NED University of Engineering and Technology, Karachi, 75270, Pakistan
10 [5] Department of Earthquake Engineering, NED University of Engineering and Technology, Karachi, 75270, Pakistan

*Correspondence to*: Haider Hasan (hhasan@neduet.edu.pk)

**Abstract.** New Makran simulations imply two generalized zones of mid-field tsunami hazard in greater Karachi. The simulations presuppose seven megathrust ruptures that strike east-west, range in area from 100x150 km to 355x800 km, and lie west of the city by no less than 100 km. The assumed seismic slip is uniform across each rupture area. The smallest 15 rupture approximates the 1945 Makran earthquake of magnitude 8.1, while the largest corresponds to a previously conjectured giant Makran earthquake of magnitude 9.2. None of the sources include a complication in 1945: late-arriving waves from submarine slides or splay faulting. Consequently, the first simulated wave is the largest in each of the seven scenarios. And because the sources are to the west, the simulated waves are higher, and arrive sooner, at Karachi Port (1.5 hr) than 30 km farther east at Port Qasim (nearly 3 hr). These combinations of height and arrival time can be generalized as 20 properties of two hazard zones: a western one that includes Karachi Port, and an eastern one that includes Port Qasim. The simulated flooding extends farthest inland into low-lying residential areas of the western zone. Neither hazard zone is near-field or far-field. That is, neither is near enough to the fault ruptures for felt seismic shaking to dependably warn of a fast-arriving tsunami, yet neither is distant enough to receive more than three hours of advance notice through tsunami warning systems. Our simulations are intended to support emergency management in this mid-field setting.

## 1 Introduction

Tsunamis represent a major hazard for coastal communities around the world, especially those located in the near-field zone. While cities in the near-field zone may receive preliminary indications of tsunami threats through seismic activity, and those in the far-field zone benefit from official tsunami declarations, many cities find themselves in a critical mid-field position. As a result, the local population may not always perceive the threat of an approaching tsunami, even with seismic 30 indications. Additionally, official declarations may arrive too late for mid-field cities, further complicating timely disaster response. .Examples of mid-field locations are tsunami arrival times from events such as the 26th December 2004 at Phuket





in Thailand (Tsuji et al., 2006) and the 11th December 20011 at Tokyo Bay in Japan (Sasaki et al., 2012). From tide gauges at various locations in Thailand, in Tsuji et al., it can be inferred that initially within the first 30 to 60 mins, there was a recession in the sea level before the wave came in, whereas from gauge records in Tokyo bay wave arrival between 60 and

120 mins, are seen in Sasaki et al., 2012, p.18. An example of modelling studies would be the results at Puget sound, which include the cities of Seattle, Tacoma, Olympia, and Everett along the coast of Washington State (Dolcimascolo et al., 2021).

The Makran coastline, extending across both Iran and Pakistan faces a significant tsunami threat from the Makran Subduction Zone (MSZ), where seismic activity and submarine slides contribute to the generation of near-field waves (see Fig. 1(a) and (b)). Two notable events, the 8.1 magnitude earthquake in 1945 offshore of Pasni (Byrne et al., 1992) and the

more recent 7.7 magnitude onshore earthquake (Hoffmann et al., 2014) both generated tsunamis (see locations in of earthquake in Fig. 1(b)). The 1945 tsunami resulted in approximately 300 casualties (Ambraseys and Melville, 1982; Atwater et al., 2021, p.2; Hoffmann et al., 2013) whereas the 2013 tsunami was detected along the east coast of Oman, hypothesised as being generated from a submarine slide (Hoffmann et al., 2014).

The cities of Jask, Gwadar, Pasni and Ormara along the Makran coast directly face the MSZ and are situated in the

nearshore zone. However, cities like Karachi, Muscat, or Bombay find themselves in a mid-field position. This paper seeks to address the specific hazard assessment needs of mid-field locations, with a focus on greater Karachi, amidst the potential risks posed by the MSZ. By investigating uncommonly large rupture scenarios, ranging from an approximation of the historical earthquake of 1945 (Byrne et al.) to hypothetical earthquakes of magnitudes 8.7 – 9.2 proposed by Smith et al. (2013), the study aims to evaluate the tsunami hazards faced by ports and residential areas in greater Karachi through

numerical modelling. It should be noted an earlier study by McCaffrey (2008) suggests that present evidence cannot rule out the possibility of any subduction zone producing a magnitude 9 earthquake. Additionally, unexpected events of the 2004 Sumatra-Andaman earthquake and 2011 Tohoku earthquakes unseen at these locations, underscores the importance of preparedness. Hence, a large rupture along the Makran Subduction Zone could be catastrophic along the coastline of Pakistan including the port city of Karachi whose exposure to risk has grown many folds since 1945 events (see Atwater et

al., 2021, p. 39).

The rest of the paper is structured as follows: In Sect. 2, we present an overview of the Makran Subduction Zone, including the tsunami sources, and review existing modelling studies within this region. Additionally, this section discusses the relevance of the study to the tsunami early warning system in Pakistan. Section 3 details the numerical model setup, and the earthquake scenarios selected.  Next Sects. 4 and 5 present a detailed discussion of the results, which include wave

heights and arrival times and the offshore flooding at Karachi. This is followed by Sect. 6 discussing the implications to emergency management. Finally, Sect. 7 concludes with findings form this study.

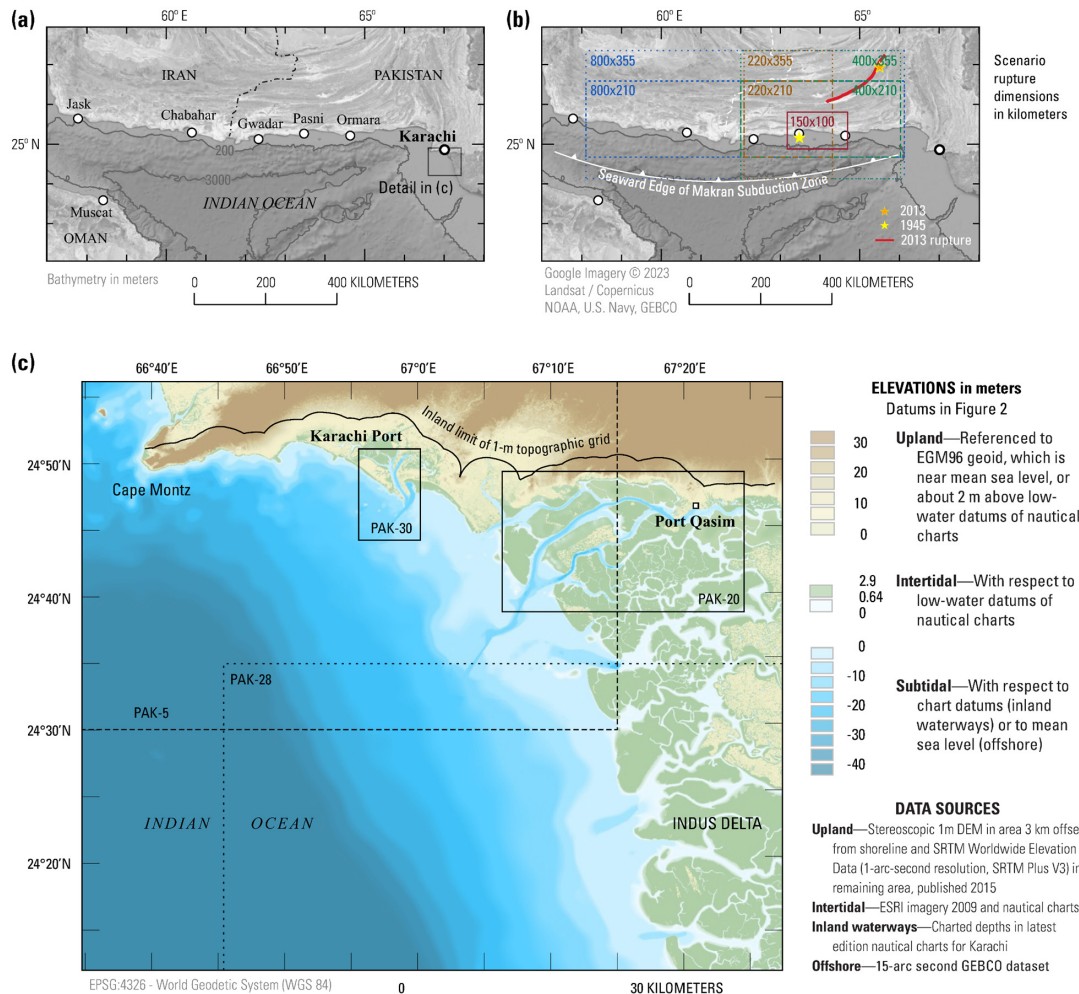

**Figure 1: (a)** Geographical Overview: Arabian Sea and surrounding nations **(b)** Dimensions and locations of hypothetical rupture scenarios: Makran Subduction Zone **(c)** Digital elevation model (DEM).



## 2 Makran Subduction Zone

### 2.1 Tsunami sources

Located in the Arabian Sea, the Makran Subduction Zone lies offshore of the coast of Pakistan and Iran. The zone starts 50 - 100 km west of Karachi and stretches westwards to almost 800 - 900 km. Here the Arabian Plate subducts northwards under the Eurasian plate at a rate of ~4 m/century (DeMets et al., 1990). The Great earthquake of Makran, with magnitude of 8.1 and ensuring tsunami, stood as the only documented event from the Makran subduction thrust. Prior to the Boxing Day earthquake ($M_w$=9.1) in 2004 off the coast of Sumatra, Indonesia, and resulting tsunami, the Makran tsunami was considered

to be the most destructive event in the Indian Ocean. The Makran earthquake centred in the shallow waters offshore of Pasni struck in the early hours of November 28, 1945 (November 27, 1945, at 2156:55.2 UTC). The event was just not damaging in terms of the earthquake to nearby towns of Pasni, Gwadar and Ormara (Ambraseys and Melville, 1982) but also devastation from the ensuing tsunami was felt in Karachi, Bombay (now Mumbai) and as far away as Port Victoria Seychelles (Beer and Stagg, 1946; Pendse, 1946). A comprehensive review on the impacts of the 1945 Makran tsunami can

be found in Hoffmann et al. (2013).

A comprehensive analysis of the 1945 Makran earthquake was conducted by Byrne et al. (1992) to determine its seismic parameters. Through body waveform inversions and first motions of P to determine, the earthquake was identified as having an interplate mechanism, occurring at the shallowest part of the plate interface. The fault rupture area was determined to extend beneath Ormara and Pasni, which was inferred to be 100-150 km along a plane dipping 7 degrees north.

Furthermore, the study highlighted the unique features of the Makran subduction zone, including segmentation, aseismic conditions, and differences between its eastern and western segments. The eastern segment was found to have experienced historic large earthquakes with a well-defined seismic front localized near the coast, whereas the western segment lacked clear records of such events, indicating significant segmentation within the zone. The presence of an extensive forearc in the Makran subduction zone was noted, likely contributing to the observed segmentation. The analysis also revealed that the

presence of a substantial sediment volume in the region does not necessarily indicate a low potential for thrust earthquakes along the Makran subduction zone, which was contrary to the common belief at the time and no longer widely held.

The unexpectedly devastating tsunamigenic events of 2004 Indian Ocean earthquake and 2011 Tohoku earthquake served as a wake-up call for seismic risk assessment and the understanding of whether subduction zones such as the Makran subduction zone has the potential for large earthquakes. Smith et al. (2013) focused on the Makran subduction zone and

proposed that it has the potential for an entire-length rupture, capable of generating an earthquake of magnitude $M_w$ 9+. The conclusion was drawn from uncommonly great rupture widths through heat flow modelling. The main concern was understanding where faults may break, as this determines seismic energy radiation and tectonic deformation. This is crucial for tsunami initiation. It is worth noting that the study highlighted the Makran subduction zone's seismic potential may have been underestimated. The 2D thermal model revealed high plate boundary temperatures due to sediment cover, making the




megathrust potentially seismogenic to a shallow depth. The shallow dip of the subduction plate creates a wide potential seismogenic zone.

The implications of the study included the consideration of three rupture scenarios depicted in Fig. 1(b), ranging from minimum to maximum width. These scenarios encompassed various segments of the subduction zone: the longest calculated rupture encompassing the full length of the subduction zone (~800 km), a scenario covering the Pakistan section

of the Makran (400 km), and a scenario accounting for potential impediments to rupture propagation (220 km). These scenarios produced earthquakes with a potential magnitude range of Mw 8.7–9.2. The proposed fault rupture widths, range from 210 km to 355 km, dwarf the 1945 rupture width of 100-150 km. Slip estimates from the 1945 Makran according to Byrne et al. (1992) was 7m whereas Smith et al. determined 10 m for the slip by considering a 250 year recurrence interval as the Makran convergence rate is 4cm/yr, and found that it was sufficient for magnitude 9 earthquakes at Makran because of

the great fault widths.

A more recent study has delved into the behaviour of the megathrust in the eastern Makran subduction zone. The observed triggered aseismic slip, using InSAR time series data, as detailed in Lv et al. (2022), indicates that approximately 80 cm of aseismic slip along a 5,500-km$^2$-wide subhorizontal patch of the megathrust fault occurred due to the shear stress imparted by the 2013 $M_w$ 7.7 Balochistan earthquake. This observation aligns with the conclusion that the megathrust is

considered to be fully locked to at most 220 km distance from the trench. Furthermore, the implications for the seismic potential of the subduction zone are significant. The lack of historically large earthquakes, including the absence of $M_w \geq 9$ earthquakes in the historic record, is aligned with the observed fully locked or partially decoupled behavior of the megathrust.

Thingbaijam et al. (2017) introduced empirical scaling laws for earthquake rupture geometry, focusing on

subduction-interface earthquakes. The study emphasized that such events exhibit significantly larger rupture widths and areas compared to other faulting regimes. These scaling relationships align closely with self-similar scaling, suggesting that the observed wide fault rupture widths in the Makran subduction zone are consistent with the characteristics of subduction-interface earthquakes. Moreover, Thingbaijam et al. addressed the potential constraints imposed by finite seismogenic depth on rupture width in subduction events. They proposed adjustments in the scaling relationship between magnitude and rupture

width, which emphasize higher confidence in width and the logarithm of rupture area. This adjustment reinforces Smith et al.'s conclusions regarding the seismic potential of the Makran subduction zone, as it provides a theoretical framework supporting the observed wide fault rupture widths.

Other than tectonics sources, splay faults and submarine slides are also considered to be potential sources of tsunami hazard along the Makran subduction zone (Rashidi et al., 2020a). For instance the delayed arrival time and the large

run-ups of the wave after the main shock from the 1945 event (Byrne et al., 1992) are still unexplained and have been hypothesised to be from secondary submarine landslide source (Heidarzadeh et al., 2008a; Rajendran et al., 2008; Rastgoftar and Soltanpour, 2016). The delayed arrival and magnitude of the highest wave in the 1945 Karachi tsunami investigated by Atwater et al., 2021, showed complications with the tide gauge, including stilling blockage and a mechanical outage, as





contributing factors (Atwater et al., p.16–18). Table A1 tabulates the arrival time delays of tsunami waves at Pasni, Karachi,
Mumbai (formerly Bombay), and Seychelles. The implications of the late arrival of the waves for emergency management
discussed in Sect. 6.2. Moreover, a submarine landslide source has also been hypothesized as being the generation
mechanism of the tsunami in the northwestern Indian Ocean following 24th September 2013, Mw 7.7 strike-slip earthquake
originating far inland in Pakistan (Heidarzadeh and Satake, 2014b). However, a study by Vita-Finzi (2014) attributes the
origin from a fault within the Makran accretionary prism.

**2.2 Tsunami modelling**

The significant growth in population and urbanization of the Northwest Indian Ocean shore, which includes the coastal belt
of Pakistan, has led studies to model its tsunami hazard, notably spurred by the 2004 Boxing Day tsunami and the 2011
Tohoku tsunami. These studies, categorized into deterministic tsunami hazard assessment (DTHA) and probabilistic tsunami
hazard assessment (PTHA), aim to assess the potential risks posed by tsunamis to coastal communities. A summary of the
findings of these studies is presented in Table A2.

Earlier studies in the Makran region were based on DTHA. These investigations focused on either the 1945 source
(Heidarzadeh et al., 2008a, b; Jaiswal et al., 2009; Neetu et al., 2011; Rajendran et al., 2013) or specific earthquake
scenarios, including worst-case ones by assuming single source earthquake with maximum magnitude (Heidarzadeh et al.,
2007, 2008b, 2009a, b).

The deterministic studies, particularly those investigating effects from maximum magnitude and worst-case
scenarios like those conducted by Heidarzadeh et al. (2008b) and Heidarzadeh et al. (2009a), offer crucial insights into
tsunami hazards along the Northwest Indian Ocean shore, including the coastal belt of Pakistan. Heidarzadeh et al. (2008b)
explored into a maximum regional earthquake with a magnitude of 8.3 having a 1000-year return period, determined by
seismic hazard analysis. Six distinct $M_w$ 8.3 scenarios were placed at different locations along the Main Makran Subduction
Zone (MSZ) resulting in a range of tsunami wave heights, with peaks reaching 4 to 9.6 meters along the southern coasts of
Iran and Pakistan. Furthermore, Heidarzadeh et al. (2009a) expanded on these findings by considering worst-case scenarios,
including earthquakes with magnitudes of 8.6 and 9.0, featuring rupture lengths of 500 km and 900 km, respectively. In these
scenarios, splay faults branching from the main plate boundary were integrated, adding complexity to the fault dynamics.
The analysis resulted in heightened tsunami impacts, with runup heights ranging from 12 to 18 meters for the first scenario
and 24 to 30 meters for the second scenario. The presence of splay faults was observed to amplify wave heights and local
run-up by a factor of two. Importantly, these studies were conducted post the 2004 Boxing Day tsunami but pre-Tohoku
tsunami of 2011, thus, were conceived before Smith et al. (2013) who proposed the entire rupture of the Makran Subduction
Zone. Rashidi et al. (2018a) analysed the tsunami wave energy distribution from both static and dynamic sea-floor
deformation due to a full rupture having a 210 km width and 10 m slip following Smith et al..

The limited availability of seismotectonic data for the Makran region, along with gaps in historical and paleo-
earthquake studies, recognized the necessity for a more comprehensive probabilistic approach. A first generation PTHA for a





single moment magnitude was conducted by Heidarzadeh and Kijko (2011) to determine the probability of exceedance of moderate and extreme tsunami wave heights. The study was followed by Hoechner et al. (2016) which captured the tsunami hazard using PTHA from a full spectrum of possible earthquakes for the coastlines of Iran, Pakistan and Oman. The focus of

the investigation was on significantly larger magnitude earthquakes following the Schellart and Rawlinson (2013) which concluded that the Makran subduction zone can generate events larger than $M_w$ 8.5. A more recent study by Gopinathan et al. (2021) utilizes statistical emulation to model the tsunami impact, using 300 training simulations to construct an emulator to generate 1 million predictions around Karachi Port.

The studies related to modelling the 1945 tsunami by utilizing the earthquake as the sole source, have failed to
describe the delay at Pasni and Karachi (Rajendran et al., 2013) and run-up heights at Pasni (Heidarzadeh et al., 2008a). Heidarzadeh and Satake (2014) proposed an alternative source than the one determined by Byrne et al. (1992). The author's proposed a four segmented source with varying slip amounts making use of sea level observations from 1945 published in Neetu et al. (2011). Another study by Heidarzadeh and Satake (2017) combined an earthquake-landslide model for the 1945 scenario to reproduce the 10-12 m runup observtaions during 1945.

An in-depth analysis of the delayed arrival and magnitude of the highest wave observed during the 1945 tsunami in Karachi harbour was conducted by Atwater et al. (2021). Attributed to the complications with the Karachi tide gauge, the ten-hour section of the Karachi marigram and its detided version (Adams et al., 2018) is used as the source to understand the inconsistencies behind the lag of water level at the time of the earthquake and the gap in the marigram which results in the largest wave not being recorded. The gap in marigram could possibly be responsible for the flooding of the oil facilities
(Atwater et al., p.22).

The study finds that it is the blockage of the stilling that may have been responsible for the fall in the water in level in stilling well. The blockage is explained through historical documentation which details the persistent challenges with sediment infilling and mollusk blockage (Atwater et al., p.16).  Furthermore, a mechanical outage termed as "belt of the wheel" during the tsunami event, occurring shortly after 8:00 am IST, resulted in the gap in the gauge record, thereby failing
to capture the largest wave. at 8:15am IST (Atwater et al., p.18).

**2.3 Tsunami Early Warning**

In the aftermath of the catastrophic trans-Indian Ocean Tsunami in 2004 and the unprecedented Kashmir Earthquake in 2005, Pakistan recognized the need for proactive disaster risk management. The Pakistan Meteorological Department (PMD) took significant steps by establishing a state-of-the-art National Seismic Monitoring and Tsunami Early Warning Centre
(NSM & TWEC) in Karachi with a backup centre in Islamabad with the support by the Government of Pakistan and UNESCO / IOC (PMD, 2010). The National Tsunami Warning Centre (NTWC) is responsible for providing timely and clear tsunami warnings in the event of a major earthquake occurring under the sea at a shallow depth. To achieve this, the NTWC monitors seismic activity using both national and global seismographic networks in real-time. The protocols of the centre also includes receiving tsunami advisories from international organisations such as the Pacific Tsunami Warning Centre





(PTWC) and Japan Meteorological Agency (JMA) for the Indian Ocean. Once all the necessary information is received and evaluated, the NTWC issues Tsunami Bulletins to Emergency Response Authorities and the media, outlining the potential threat to coastal areas of Pakistan.

In a workshop held in Karachi, Pakistan in February 2020, focused on refining Standard Operating Procedures (SOPs) for Tsunami Early Warning Systems (Spahn, 2020). Pakistan, situated in the near-field zone for tsunami generation,
shares similarities with Oman in terms of warning timeframes. It was suggested that the initial warning message in Pakistan should reach the public within 7-12 minutes after the earthquake. This recommendation underscores the urgency required for swift communication to enable timely evacuation and preparedness measures. However, for Karachi, which is situated in the mid-field zone for tsunami generation, the tsunami wave is expected to arrive within 1 to 2 hours after the earthquake. The report by Spahn, mentions another protocol forwarded for Oman, where by warning is issued to the public within 10-90 mins
for events other than near-field, which could serve as a model for Karachi.

**3 Numerical model setup**

Tsunami hazard for Karachi is simulated using GeoClaw, an open source code based on the Clawpack (Conservation Laws Package) software (Clawpack Development Team, 2020). GeoClaw has been widely used to model hazards from geophysical flow phenomena such as tsunamis (Garrison-Laney et al., 2021), storm surge (Mandli and Dawson, 2014; Toyoda et al., 2022) and landslide generated outburst (Turzewski et al., 2019). Approved by the US National Tsunami
Hazard Mitigation Program (NHTMP), GeoClaw employs a high-resolution shock capturing numerical finite volume method together with adaptive mesh refinement (AMR) on spherical grids (Berger et al., 2011). For accurate simulations, the code incorporates the Okada model (Okada, 1985) to simulate vertical coseismic deformation, and factors in Coriolis forcing, bed friction through the Manning coefficient, spatially varying sea levels, and can also read multiple overlapping data sets of
complex realistic topography and bathymetry having different resolutions.

**3.1 Digital elevation model (DEM)**

A digital elevation model (DEM) for Karachi (See Fig. 1(c)) was developed that represented the latest topographic and bathymetric features. Bathymetric soundings nearshore were used from Nautical Charts available from Pakistan Navy (Bhatti, 2011, 2012, 2017a, b) and for off-shore regions, the GEBCO data from 2020 where bathymetric data (GEBCO
Bathymetric Compilation Group 2020) is utilized. Tidal shorelines were extracted from ESRI imagery (ESRI World Imagery, 2021) with reference to Nautical Charts.

Vertical datums referencing mean sea level (MSL) were established for Karachi Port and Port Qasim, facilitating the alignment of depth data. Figure 2 illustrates the vertical datums at the tide gauges for the two ports including the datum in 1945 at the old tide gauge location in Karachi. Notably, a sea level rise of 21 cm from 1992 to 2018 was observed,

Natural Hazards and Earth System Sciences
Author(s) 2024
en





attributed to climate change-induced ice melt. Hogarth (2014) also documented a similar number for sea level rise at Karachi since 1940s.

The DEM includes upland topography represented by a gridded data of the Shuttle Radar Topography Mission (SRTM) having a spatial resolution of 1-arc second or 30m (Nikolakopoulos et al., 2006a), however, 3 km offset from the shoreline, a 1m high-resolution DEM model developed from stereoscopic orthoimages (Digital Globe, 2016) is used. The

final DEM has a 10m resolution offers a comprehensive representation of Karachi's topography and bathymetry, crucial for various applications including tsunami modelling. For details see Appendix B.

In this study, constant ambient tidal scenarios are considered, with sea level variations representing historical and potential future conditions. Specifically, three different constant ambient tidal scenarios are examined: (1) A sea level 0.2 meters above Mean Sea Level (MSL), which approximately corresponds to reported tide levels at Karachi port during the

1945 Makran Tsunami; (2) sea levels based on charted highest astronomical tide (HAT) values, with 1.7 meters above sea level at Karachi Port and 2.29 meters above sea level at Port Qasim; (3) The third scenario, representing a worst-case scenario, considers the HAT at Port Qasim, which could possibly mimic future sea level rise under climate change impacts. These scenarios provide insights into the potential impact of varying sea levels on tsunami hazards in the Karachi region.

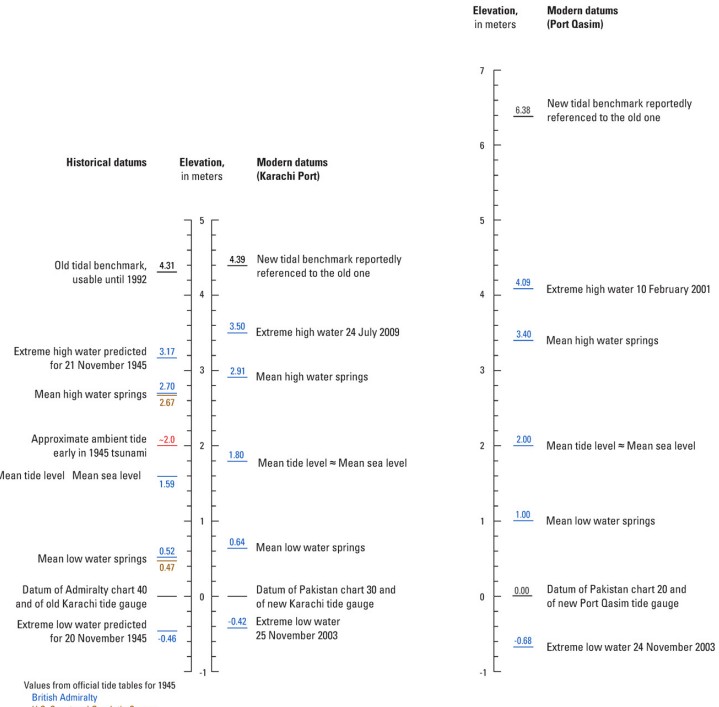

**Figure 2:** Vertical datums for Karachi Harbour and Port Qasim. References and other details in the text.





### 3.2 Scenarios selected

Following the discussion of the tectonics for the Makran subduction zone in Sect. 2.1, we select seven hypothetical rupture scenarios to investigate the tsunami hazard along the shore of Karachi. While Lv et al. (2022) suggests that the section of the megathrust fault is locked at a maximum distance of 220 km from the trench, the earlier paper by Smith et al. (2013) introduces a 2-D thermal model of the Makran subduction zone that highlights wider potential seismogenic behavior. The high temperatures at the plate boundary, resulting from the thick sediment cover on the incoming Arabian plate and the

shallow dip of the subducting plate, contribute to a wider potential seismogenic zone of up to ~350 km. By considering a width of 350 km for the megathrust fault, we aim to encompass a broader spectrum of possible rupture scenarios. However, it is crucial to acknowledge that the findings of Lv et al. may cast doubt on the downdip extent proposed by Smith et al.. The choice to retain the 355-km rupture width is supported by consideration of the updip limit and its potential for producing more deformation in deep water compared to scenarios limited to 220 km. While the downdip limit far inland may not

significantly influence this decision, the observed indicators of aseismic slip in 2013 are deemed to outweigh the results of heat-flow models and assumptions about the temperatures at which the Makran megathrust may transition from ductile to brittle.

Fig. 1(b) shows the rupture locations and rupture dimensions of these ruptures. The rupture area bounded in red, is the source area of the great 1945 Makran earthquake inferred from Byrne et al. (1992). The rest of the six hypothetical

scenarios are based after Smith et al. (2013); the dimension for these rupture areas are illustrated in the figure represented by the rest of the colours. The ruptures vary in length and width: 1. long wide (800x355) and long narrow (800x210); 2. medium wide (400x355) and medium narrow (400x210); and 3. small wide (220x355) and small narrow (220x210). It should be noted that for simplicity the rupture areas assumed to be rectangular.

The seafloor deformation for the rupture scenarios are shown in Fig. 3. These are determined using the Okada

model with the associated earthquake parameters listed in Table A3. For the Byrne et al. representation of the rupture area (150x100 km) a uniform slip of 7 m is considered. The uplifts seen in seen Fig. 3(a) are mainly offshore of Pasni and Ormara with the maximum between 2 and 3 meters. For the rest of the scenarios a 10 m slip is considered, and the vertical seafloor deformation can be seen in Fig. 3(b) which as large as 4 m. These uplifts are mainly offshore and as seen from Fig. 4, a cross-sectional profile shows the extent of how far offshore the uplifts are with respect to the width of the rupture area. For

the wide ruptures of 355 km extending all the way inland and offshore towards the ocean floor.

Some other simplifying assumptions that have been considered in this study are that no secondary sources are considered such as slides and splay faults.

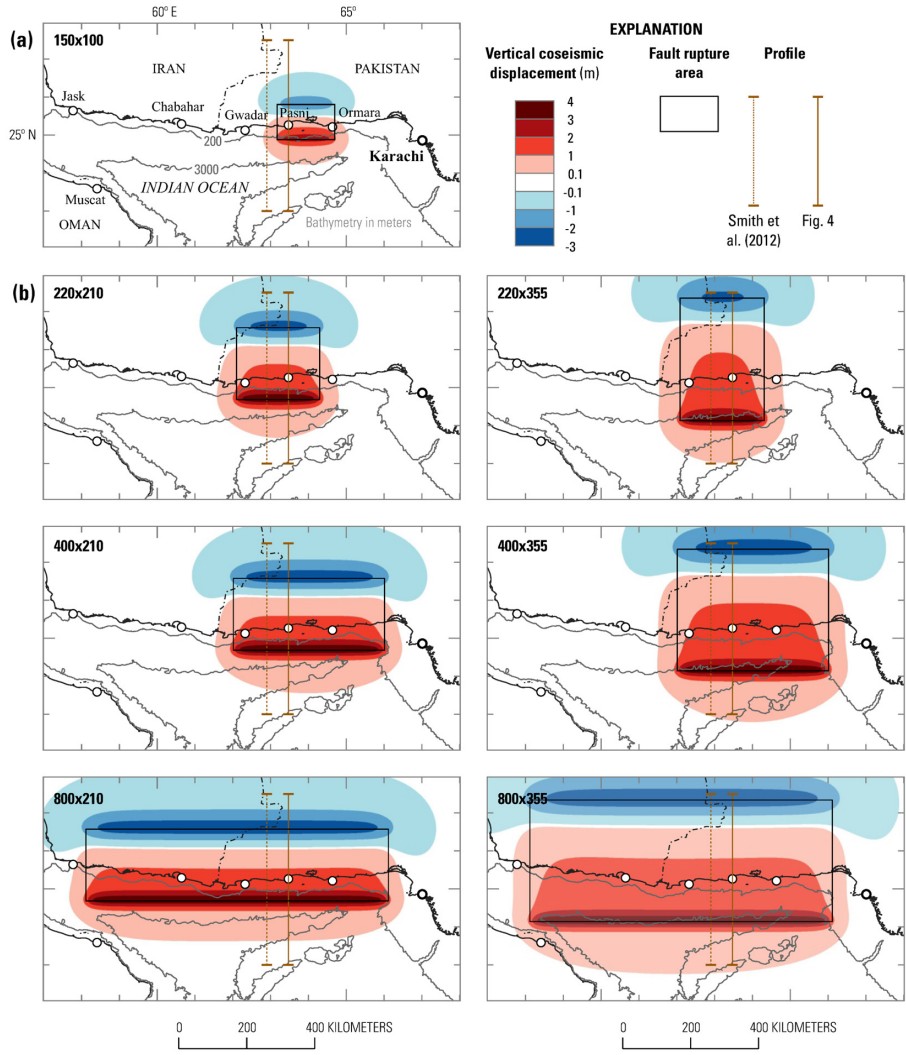

**Figure 3:** Vertical coseismic seafloor displacement of hypothetical plate-boundary ruptures.




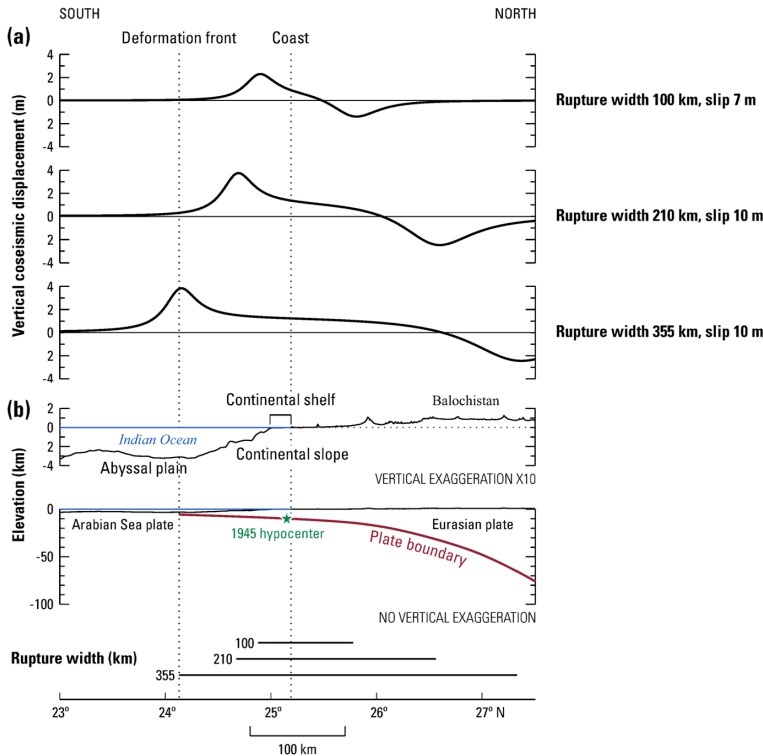

**Figure 4:** Cross-sectional profile of the hypothetical rupture in Fig. 3

## 4 Wave heights and arrival times

We simulate the tsunami waves triggered by the seven hypothetical rupture scenarios discussed in Sect. 3.2. Each of the scenarios are simulated for 24 hrs and their results are compared on a regional level at various cities dotted along the North West India Ocean and as far south as Port Victoria, Seychelles (see Fig. 5(a)). Sea level fluctuations were observed at Port

Victoria during the 1945 event (Beer and Stagg, 1946). Wave heights and arrival times are also compared on a local level at various locations along the entire coast of Karachi, from Cape Monze in the East to Russian beach further west of Port Qasim (see Fig. 6). The simulations make use of the adaptive mesh refinement (AMR) feature in GeoClaw to efficiently model the generation, propagation, and inundation of the tsunami. At the regional level, the finest grids were resolved to around 230 m, however, to model the inundation at Karachi, the finest level for grid resolution was 10 m.





### 4.1 Indian Ocean

On the regional level, the tsunami wave for the seven hypothetical rupture scenarios were analysed at the synthetic tide gauges placed along the coast of the administrative boundaries of the various cities (see Fig. 5(b)). The gauges were placed at 10 m depth contour. The extracted maximum wave heights, the associated arrival times, and uplifts along the coast of the various cities are illustrated in Fig. 5(a), the values of which are tabulated in Table S3. Evident from these plots is that as the waves radiate outwards from the source, the wave heights decrease, and the associated arrival times increase thus revealing a clear relationship between the source distance and the resulting wave height and arrival time.

*Scenario 150x100*

The rupture area for the 1945 earthquake encompassed the cities of Pasni and Ormara. Coastal locations, such as Pasni and Ormara, closer to the earthquake source experienced more substantial wave heights and shorter arrival times. Pasni, experienced a wave height of 2.1 m above mean sea level (MSL), while Ormara had a peak of 1.92 m. These waves arrived rapidly, with Pasni, recording an arrival time of 32 mins after the event. It is worth noting that this rapid arrival time contradicts eyewitness accounts who reported the wave arriving 1.5 to 2 hrs after the earthquake (Page et al., 1979). Uplifts associated with these wave heights are 0.8519 m at Pasni and 0.8905 m at Ormara. However, it is crucial to acknowledge the discrepancies between simulations and historical observations, especially the absence of observed uplift at Pasni in contrast to the reported 2 m uplift at Ormara (Page et al., 1979).

Moving east of Ormara at Karachi and Bombay (now Mumbai), wave heights reached 0.89 m and 0.36 m, respectively with arrival times of 1 hr 42 mins, and 4 hrs 36 mins after the earthquake. Notably, the tide gauge at Karachi Port recorded the third wave as the highest, measuring 0.56 m, with an arrival time of 4 hrs and 21 mins (from Adams et al., 2018). However, it should be noted the gauge had malfunctioned and failed to record the maximum wave, which occurred around 4 hrs and 53 (Adams et al., 2018; Neetu et al., 2011). Neetu et al.'s study reported the first wave to be the highest at Bombay which was recorded as 0.34 m, with a wave arrival time of 4 hrs and 41 mins based on data from the Bombay Marigram. At Seychelles, the first wave was the most significant, with a travel time of 6hrs and 15 mins (Beer and Stagg, 1946), which closely matches our model's determination of 5 hrs and 50 mins.

The simulations, as shown in Fig. 5(a) accurately depict the variation in wave heights and arrival times along the other affected areas of the North Arabian shores. These variations align well with observations summarized in table 1 of the study conducted by (Hoffmann et al., 2013). It is important to recognize these discrepancies between simulations and historical may arise from various factors, including the inherent complexity of accurately modelling real-world events and limitations related to available data, including the placement of synthetic tide gauges. Therefore, while simulations provide valuable insights, they should be considered alongside historical accounts for a comprehensive understanding of the 1945 tsunami's impact. Additionally, it is worth noting that submarine slides were also reported in the aftermath of the earthquake





(see (Ambraseys and Melville, 1982; Page et al., 1979), which likely contributed to variations in wave heights and arrival times. A summary of the arrival times for simulation vs the reported are tabulated in Table A1.

*Scenarios adapted from Smith et al. (2013):*

The results for the rest of the rupture scenarios seen in Fig. 5(a) show notable variations in wave heights and arrival times at various locations along the coast. These can be attributed to both the rupture width (rupture area) and the proximity of the seaward edge of the each of the scenario to the Makran coastal belt. Additionally, differences between scenarios may also be influenced by the volume of water being displaced.

It is evident from the figure that large rupture areas tend to generate higher wave heights. For instance, Scenario 800x355, which represents the full-length rupture of the Makran subduction zone with maximum width, produces the highest wave heights. Gwadar experiences substantial wave heights of 7.76 meters. However, the maximum wave height in this scenario is observed at Muscat, with a wave height of 9.09 meters, accompanied by positive uplift of 0.1855 meters. Similarly, Pasni and Karachi observe wave heights of 4.58 and 3.77 meters, respectively. In contrast, Scenario 220x210, with

a smaller rupture area, results in lower wave heights, with Gwadar's wave height reduced to 4.46 meters. However, in this scenario, the maximum wave height occurs at Gwadar whereas the maximum uplift occurs at Pasni, with positive uplift of 1.23 meters, along with positive uplift in other cities.

The proximity of the seaward edge of the scenarios to the Makran coastal belt also plays a critical role in determining the wave heights and arrival times. For instance, Scenarios which have their seaward edges closer to the coastal

belt, lead to lower wave heights along the coast. In terms of arrival times, narrower rupture areas tend to produce faster arrival times due to quicker wave propagation through a smaller area particularly those locations which are located within the rupture area.

As for Seychelles, it is located significantly farther from the source of the Makran subduction zone, and its wave heights and arrival times are relatively lower and longer, respectively, with positive uplift being negligible. The impacts on

Seychelles, as a remote island nation, are less severe compared to cities closer to the source.

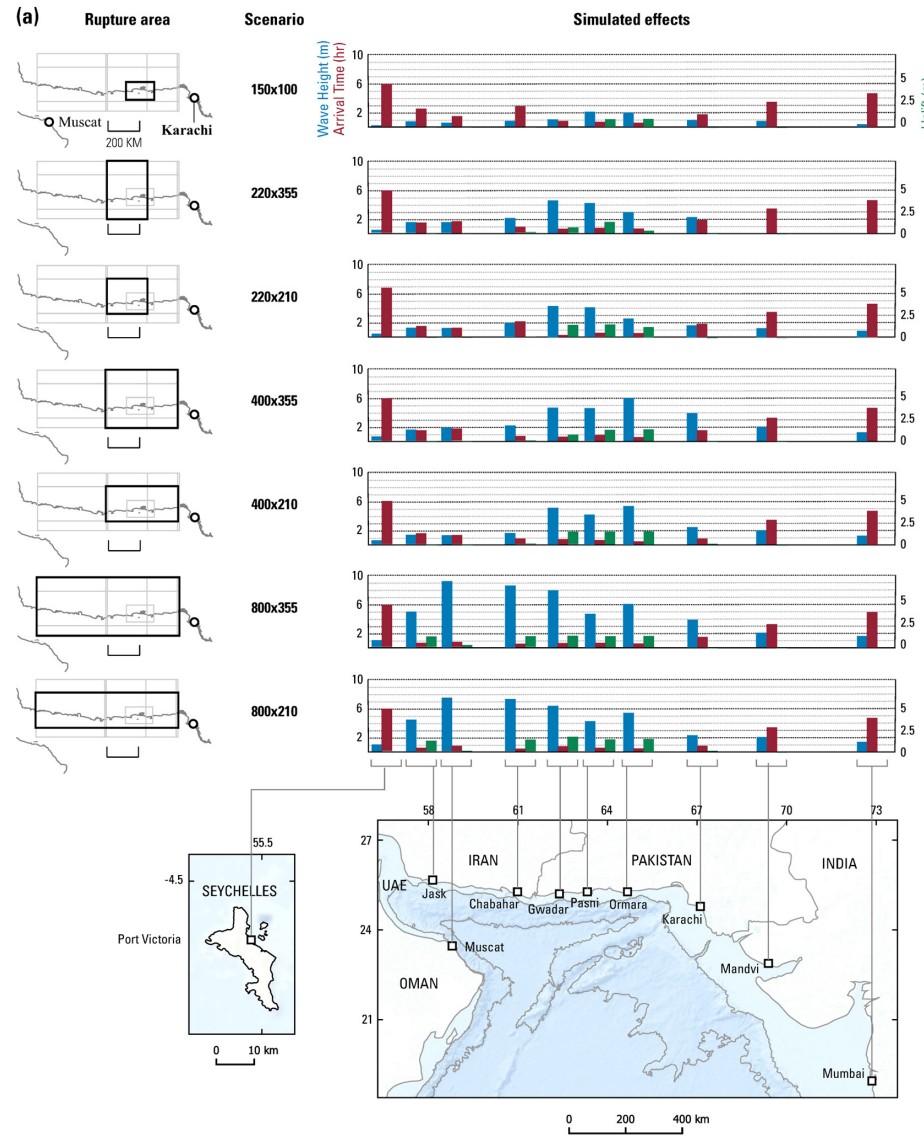

**Figure 5: (a)** Tsunami Arrival times, wave heights and uplift/subsidence at various cities of the North West Indian Ocean region including Port Victoria, Seychelles determined from the 1945 earthquake source and potential ruptures proposed by Smith et al. (2013). *(Continued on next page)*



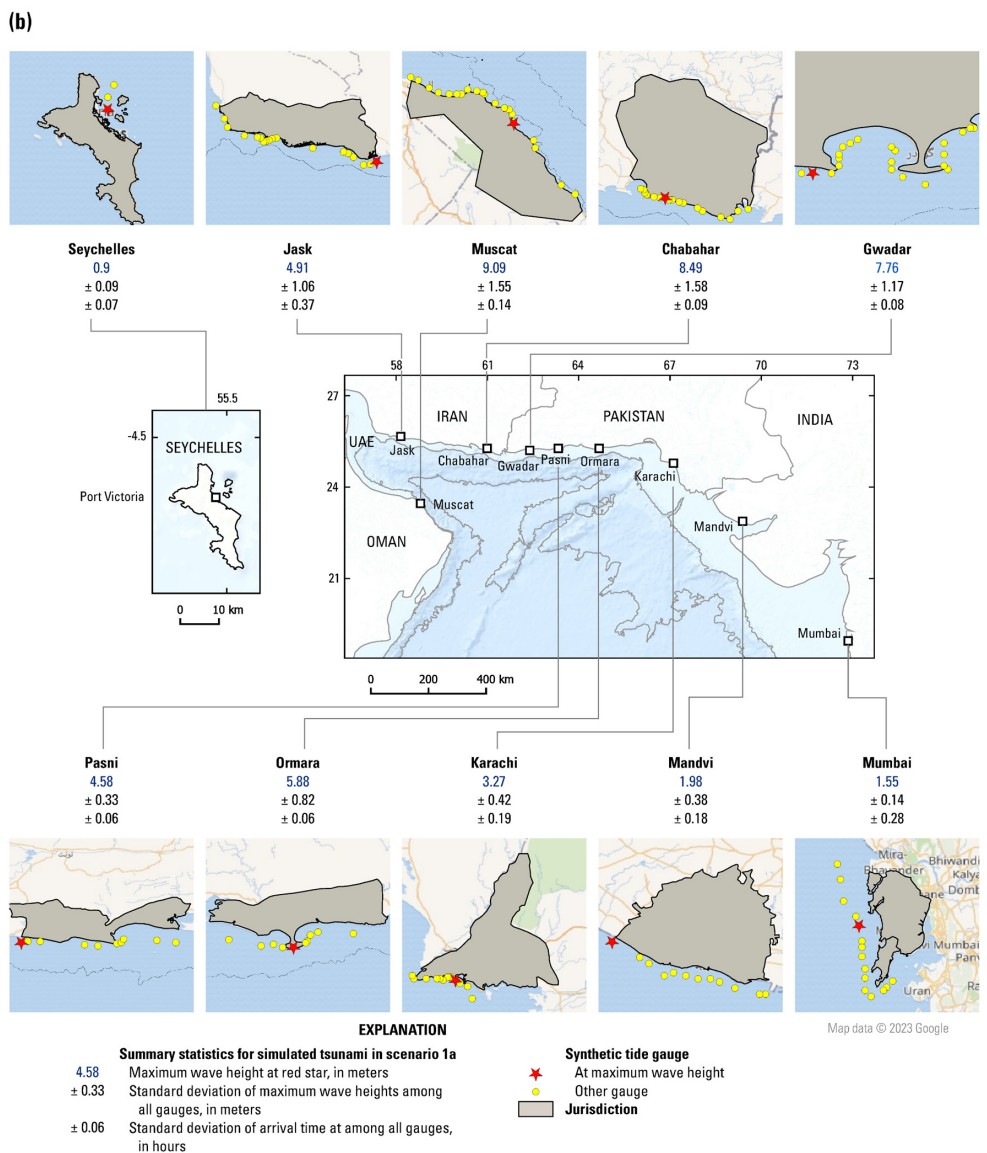

**Figure 5: (b)** Synthetic tide gauge locations along the coast of the administrative boundaries, maximum wave height and its location for scenario 800x355 and the standard deviation of wave heights and arrival times for all the gauges.




### 4.2 Greater Karachi

Findings in context to assessing the impact of various rupture scenarios on wave heights and arrival times in the Greater Karachi region are illustrated in Fig. 6, values are tabulated in Table A5. This figure provides a visual representation of the results at key locations along the coast, including Cape Monze, Manora, Rehri Goth, Lath Basti, Port Qasim, and Russian

Beach.

Cape Monze, located close to the potential tsunami source, provides valuable insights into the impact of various rupture scenarios on wave heights and arrival times, which are essential for an effective early warning system. The results at this location serve as a reference point for assessing the potential tsunami effects along the Karachi coast.

In the scenario related to 1945 (150x100), the wave height at Cape Monze is recorded at 0.77 meters, and the

tsunami arrives approximately 1.14 hours after the event. Comparing this to Scenario 210x355, notable differences emerge in terms of wave heights, but arrival times remain relatively unchanged. The wave height at Cape Monze is markedly higher, at 1.32 meters, with a slightly earlier arrival time, at 1.13 hours. The broader rupture area contributes to larger wave heights. Scenario related to 220x210, result in wave heights at Cape Monze results in waves 1.09 meters high, but with slightly later arrival time of 1.18 hours. This suggests that the width of the rupture area plays a significant role in influencing wave

heights, however, proximity of the eastern edge of the source leads to similar arrival times.

The long wide rupture (scenarios 800x355) of the Makran subduction zone leads to a wave height of 2.94 meters at Cape Monze, with an arrival time of 0.91 hours. This substantial increase in wave height and earlier arrival time can have significant implications for tsunami preparedness and response, emphasizing the importance of considering various rupture scenarios for early warning systems. Comparing these results to the long narrower rupture (800x210), the wave height at

Cape Monze is 2.01 meters, and the arrival time is similar to long wide rupture scenario. This highlights that the width of the rupture area plays a crucial role in determining wave heights, while the arrival times remain relatively consistent and depend on how far east the source is.

Moving eastwards along the coast, the trend in wave heights and arrival times is consistent across all scenarios, reflecting the influence of both the rupture area's width and geographical features. At Manora, situated at the Karachi Port,

the results are of particular importance due to its role as a strategic urban area and a port. In the 1945 scenario, wave heights at Manora measure 0.94 meters, with an arrival time of 1.63 hours post-event. For case where the rupture length 210 km but extends seaward, wave heights increase to 2.39 meters, and the tsunami arrives approximately 1.78 hours after the event. The broader rupture area causes larger wave heights, with arrival times remaining relatively early compared to locations further east. In the scenario featuring the same rupture length but a narrower rupture area, wave heights at Manora are 1.62 meters,

arriving at 1.77 hours. This demonstrates how the width of the rupture area influences wave heights, though the arrival time remains relatively early.



The long wide rupture results in wave heights of 4.20 meters at Manora, with an arrival time of 1.37 hours whereas for narrower rupture area, wave heights at Manora are recorded at 2.21 meters, with arrival times similar to wider rupture. This reiterates the impact of rupture area width on wave heights. This substantial increase in wave heights can have significant implications for port facilities and the urban area, emphasizing the need for preparedness and early warning systems tailored to such scenarios.

Moving further east along the coast, we encounter locations like Rehri Goth and Lath Basti in the Malir District, where the presence of creeks significantly reduce the wave heights. The shallower areas pose unique challenges in the context of tsunami risk and early warning systems. These areas may experience more complex wave behaviour due to the interaction between incoming tsunami waves and the coastal geography, making it essential to account for such local variations when implementing effective early warning and mitigation strategies.

Port Qasim, an industrial hub in the region, is affected differently by the rupture scenarios. The wave heights increase with respect to wider rupture, but the arrival time is considerably extended, highlighting the potential for early warning systems to offer valuable lead time for preparedness even with elevated wave heights. Russian Beach, located to the far east, records consistent results across the scenarios, with wave heights remaining relatively moderate and arrival times extending.

These variation in wave heights and arrival times along the coast underscores the need for tailored early warning systems that account for local geographical features and variations between rupture scenarios, enabling effective risk mitigation and evacuation strategies. Moreover, the substantial increase in wave heights as the rupture width increases can have significant implications for port facilities and the urban area, emphasizing the need for preparedness and early warning systems tailored to such scenarios.

### 4.3 Synthetic tide gauge recordings at major ports in Karachi

### 4.3.1 Karachi Port

Figure 7 provides crucial context regarding the historical and present-day state of Karachi Port. Figure 7(a) illustrates the apparent flow path of a tsunami towards oil facilities within the port, whereas Fig. 7(b) shows the development of the Deep Water Port, which has replaced the tidal flat, influencing the overall port landscape.

Fig. 7(c) presents the detided marigram for the 1945 tsunami, providing historical context for comparison with the simulation results. Fig. 7(d), (e), (g), and (h) shows tide gauge recordings of water surface elevation obtained from numerical simulations conducted at Karachi Port, with a focus on rupture scenarios characterized by dimensions 150x100 and 800x355. Furthermore, Fig. 7(f) presents the elevation of the oil terminal facilities, which were flooded during the 1945 tsunami.



**Figure 6:** Tsunami Arrival times and wave heights at various locations along shore of Greater Karachi region determined from the 1945 earthquake source and potential ruptures proposed by Smith et al. (2013).





The simulations consistently reveal a leading wave as the highest, in contrast to historical detided marigrams that reported the fourth wave as the most significant during the 1945 tsunami. The discrepancy is attributed to the malfunction of the tide gauge during the historical event, which does not pick up the wave height. There is alignment of arrival times of this leading wave including the first wave in for the detided marigram.

An observation from the simulations of waves shows the leading wave's increased height within the deep water port suggesting a significant water level rise, potentially leading to the flooding of the port area. Time series of water surface elevation for other scenarios at various locations of the synthetic tide gauges in and around Karachi Port are shown in Fig. C1.

### 4.3.2 Port Qasim

Figure 8 presents the time series of water surface elevations at gauge locations situated at Bundal Island, which marks the entrance to the Port Qasim Channel, and at Port Qasim itself. The focus is on rupture scenarios characterized by dimensions 150x100 and 800x355. For additional gauges within Port Qasim and its channel, refer to Fig. C2.

A notable trend observed in the data is the significant decay in wave height as they progress towards Port Qasim. Additionally, there is a noticeable delay in the arrival times, amounting to approximately 1.5 hours. This temporal shift in wave arrival carries implications for the port's vulnerability and suggests the need for careful consideration in assessing potential risks. Of particular concern is the identification of wave oscillations at Port Qasim, especially for scenarios involving large ruptures. This observation raises concerns about potential dangers posed to marine vessels navigating within the port.

## 5 Onshore flooding

### 5.1 Effects of the ambient tide

Illustrated in Fig. 9, we examine three distinct tsunami inundation scenarios based on the 400x355 rupture area along the Greater Karachi coastline yields insights into the impact of varying ambient tide levels. Contrary to Fig. 6, the 800x355 scenario is found to be the only slightly severe in terms large wave heights and earlier wave arrival times in comparison with the 400x355. However, we choose to present the 400x355 results as the western segment of the Makran subduction zone has been relatively quiet seismologically, hence, making it more relevant to the region's seismic activity.

The 400x355 rupture scenario, serving as a baseline with an ambient tide of 0.2m above mean sea level (represented by green) in Fig. 9, delineates a limited inundation footprint. Subsequent simulations, featuring ambient tide levels of 1.7m (orange) and 2.29m (red) above mean sea level, exhibit progressively broader inundation extents, highlighting the influence of heightened ambient tide levels.


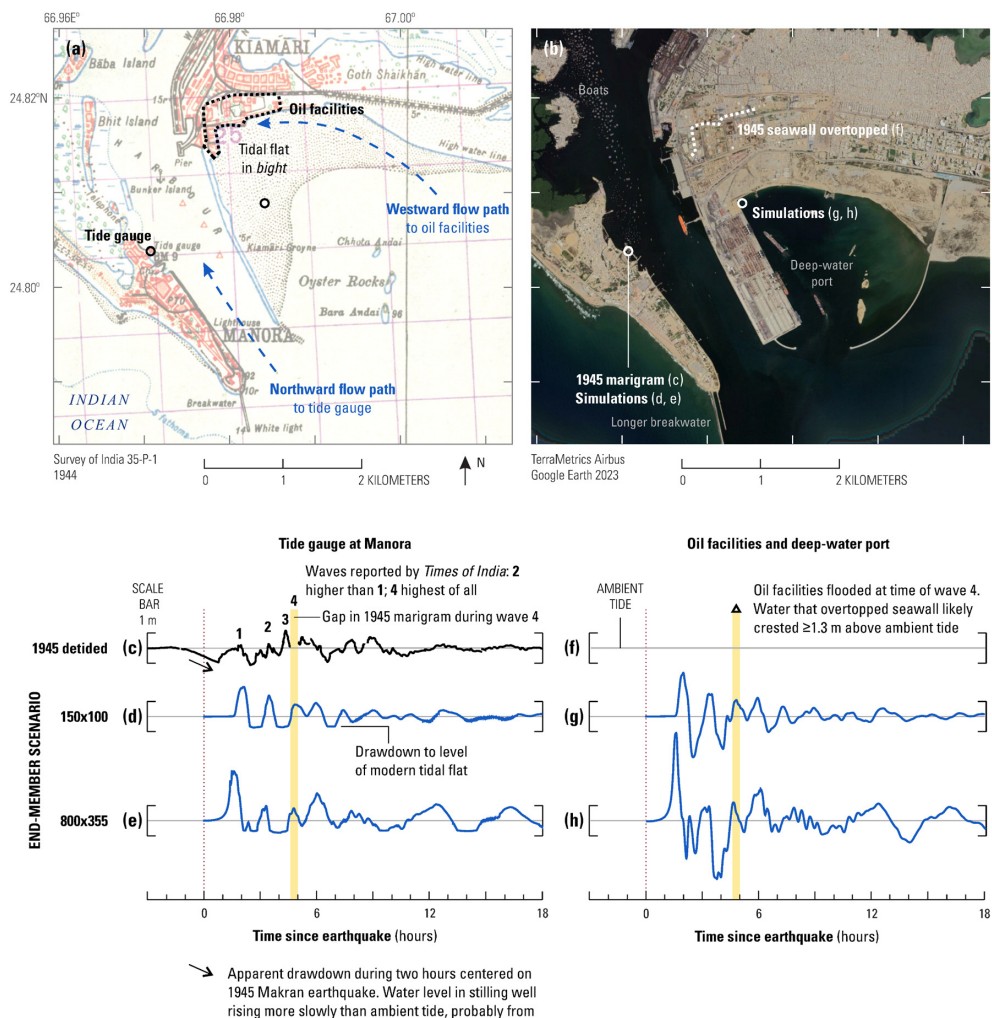

**Figure 7: (a)** Karachi Guide map showing the apparent flow path of the tsunami towards the Oil facilities within the Karachi Port **(b)** Deep Water Port **(c)** Detided Marigram for the 1945 tsunami in Karachi Port **(d) (e)** time series of the water surface elevations at the old tide gauge location for rupture scenario 150x100 and 800x355 **(f)** elevation of the oil terminal facilities **(g) (h)** time series of the water surface elevations within the deep water port for rupture scenario 150x100 and 800x355


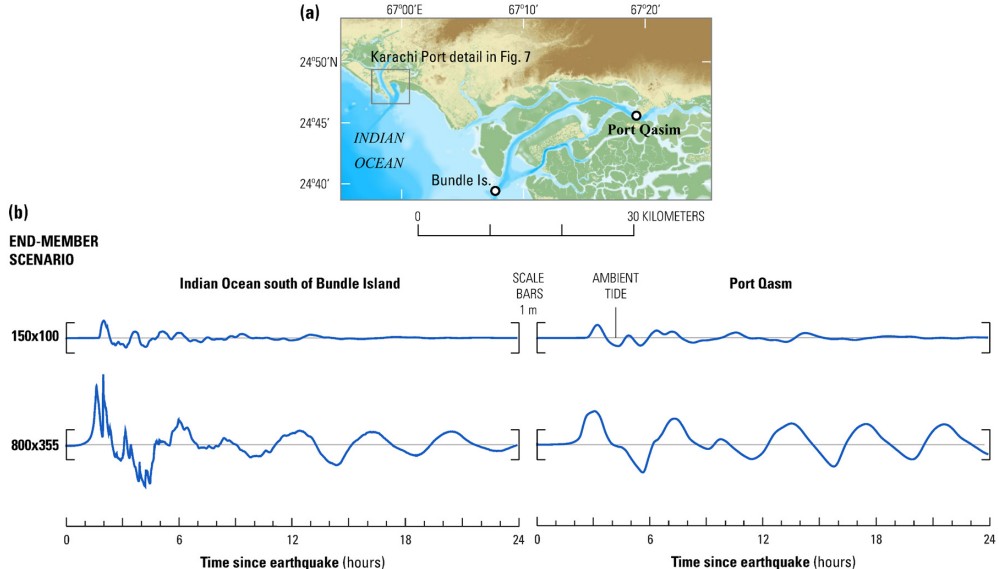

**Figure 8: (a)** Port Qasim and the approach channel **(b)** Time series of the water surface elevations at synthetic tide gauge locations at Bundal Island and Port Qasim for rupture scenario of 150x100 and 800x355

Focusing on the central region of Greater Karachi shown in Fig. 9 (b), which encompasses Karachi port, major

urban centres, and upscale residential areas, the impact of inundation becomes particularly pronounced. This region, characterized by lower topography, emerges as a focal point of vulnerability. The baseline scenario shows limited inundation, but as ambient tide levels increase, the central region experiences a more extensive and severe impact, as depicted by the broader orange and red areas, respectively.

Zooming into the central region underscores the severity of the inundation, emphasizing the potential risks to

critical infrastructure and densely populated areas. The incremental escalation in inundation extent aligns with the heightened ambient tide levels, reinforcing the importance of considering such factors in emergency preparedness and risk mitigation efforts specifically tailored for this central area.

Moving towards the east, however, the impact of inundation becomes less severe due to the protective influence of creeks along the coastline (see Fig. 9(c)). The presence of these natural features serves as a mitigating factor, shielding the

eastern regions from the full brunt of tsunami waves. This protective effect is visible in the figure, where the inundation extent diminishes as one progresses eastward. The creeks act as a natural barrier, reducing the vulnerability of the shoreline in this direction.





Furthermore, when examining the influence of higher topography on tsunami impact, the western region of Greater Karachi in Fig. 9(a) stands out as experiencing less severe inundation. The elevated terrain in this direction acts as a natural

protective barrier, mitigating the reach of tsunami waves and reducing the extent of inundation. This contrast in impact between the central and western regions emphasizes the localized nature of vulnerability, calling for nuanced risk assessment strategies that account for the diverse topographical features along the Karachi coastline.

In Fig. 9(a)-(c) we have also plotted the time series of the wave surface elevations for different ambient tides at synthetic tide gauges located at Cape Montz at the western edge of Greater Karachi, Manora in the Central region, and Port

Qasim in the east. The figure clearly shows progressive eastward delay of the waves as it approaches Port Qasim. The arrival times is just under an hour at Cape Montz and as it approaches Manora the arrival times are just above one hour. Arrival time at Port Qasim is just under 3 hours.

In addition to arrival times, these figures present insights into the impact of the tsunami travel over various ambient tide levels – 0.2m, 1.7m, and 2.9m above MSL with respect to the wave's vertical perspective as it approaches the shoreline.

Waves traveling over different ambient tide levels enable managers to gauge the potential severity of coastal inundation. For instance, in the inundation figures it is revealed that waves traveling over a 2.9m ambient tide result in inundation extending further inland. Hence, disaster responders can gauge the potential severity of coastal flooding and respond accordingly.

**5.2 Maximum flow depths**

Focusing on the Scenario with an ambient tide of 2.29m above mean sea level, which corresponds to the highest

astronomical tide (HAT) at Port Qasim, Fig. 10 illustrates a color-coded representation of tsunami flow depths. The figure serves as a visual guide for assessing and understanding the varying levels of tsunami hazards along the Karachi coast. The situation marked by regions with flow depths greater than 2.5 meters, depicted in red colour, signal a very high hazard and an imminent threat of devastating tsunami impact. In areas coloured orange, the hazard is high, exposing the area to significant risk as flow depths range from 1.5 to 2.5 meters. Of specific concern are the central areas of Karachi show in Fig.

10(b) which are the most vulnerable due to being low lying are exposed to extreme flow depths. Additionally, sporadic areas in the west and east of Karachi exhibit extreme inundation particularly in low lying areas (see Fig. 10(a) and Fig. 10(c), respectively).

This scenario serves as a crucial benchmark for understanding the maximum potential impact in the central region, aiding in the formulation of comprehensive emergency preparedness measures and risk mitigation strategies tailored to the

unique characteristics of Greater Karachi's central coastal zone. The figure further distinguishes low hazards with dark green representing minimal flow depths (0.2 meters or less), areas with slightly more noticeable effects in light green (0.2 to 0.5 meters), and zones of medium hazard denoted by yellow (0.5 to 1.5 meters).

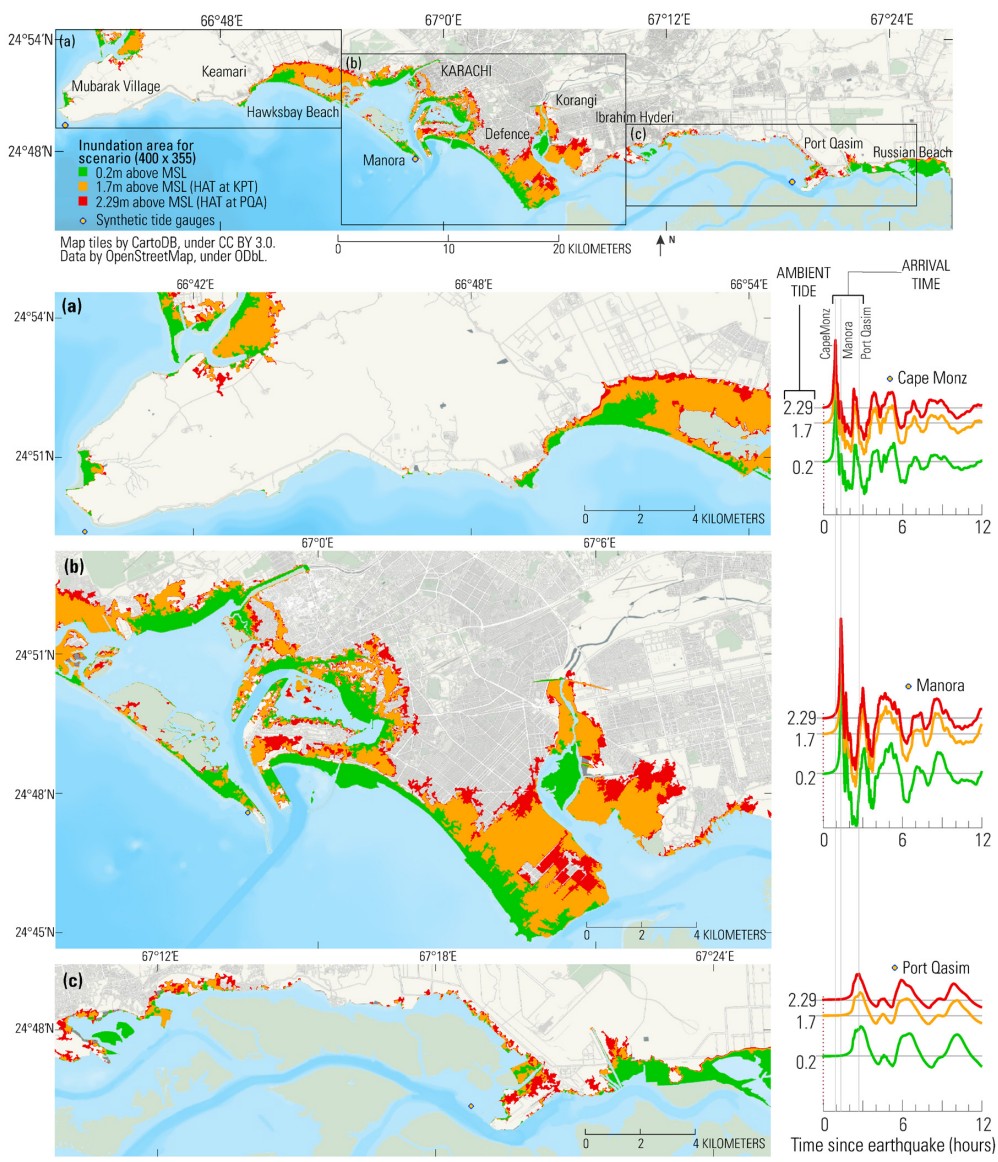

**Figure 9:** Inundation extents related to scenario 400x355 for various ambient tides 0.2m, 1.7m and 2.29m above mean sea level and corresponding time series of water surface elevations at synthetic tide gauges locations at Cape Montz, Manora, and Port Qasim.



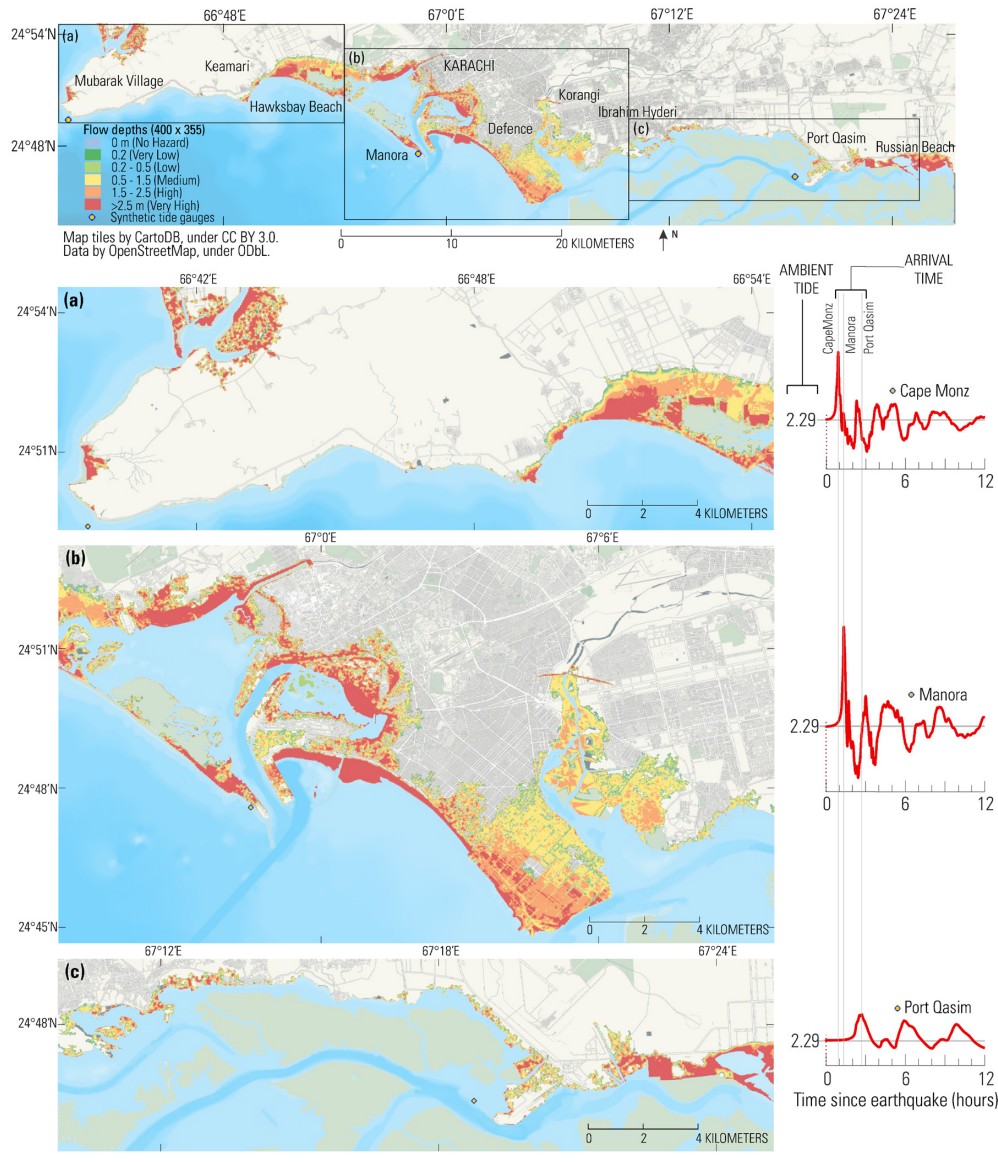

**Figure 10:** Flow depth related to scenario 400x355 for an ambient tide 2.29m above mean sea level and corresponding time series of water surface elevations at synthetic tide gauges locations at Cape Montz, Manora, and Port Qasim.





## 6 Implications for emergency management

### 6.1 Two-generalized zones of mid-field tsunami hazards

The study's findings, as illustrated in Fig. 6, reveal the delineation of two generalized zones of mid-field tsunami hazards along the coastline of Greater Karachi. The zones being the western part of Greater Karachi, including the vulnerable central region, and the eastern part of Greater Karachi, mainly comprising the coastal line of District Malir. This delineation is based on delayed arrival times and relatively low wave heights in the eastern region attributed to protection from natural features compared to the western region. Understanding and acknowledging these zones are paramount for effective emergency management and risk mitigation strategies.


*Central region vulnerability*

The central region of Greater Karachi emerges as a focal point of vulnerability due to its lower topography as evident in Fig. 9(c) and Fig. 10(c). In contrast, west part of the city which is closer to the source experiences less inundation due to higher upland elevations, as depicted in Fig. 9(b) and Fig. 10(b). Inundation scenarios reveal the central region experiencing

extensive and severe impacts, particularly during scenarios characterized by larger rupture areas and heightened ambient tide levels. This region is characterized by critical infrastructure, major urban centres, and densely populated residential areas which will be at a potential risk during a tsunami.

To effectively mitigate tsunami risks effective evacuation plans needs to be devised based upon designated safe zones. Plans should not only outline evacuation routes but also incorporate communication strategies to ensure timely and

orderly evacuation. This will provide necessary information to reorganize and reallocate limited emergency supplies available within the hazard zones.

The critical infrastructure warrants a comprehensive risk mitigation strategy which may include structural rehabilitation of public facilities and associated infrastructure. Consequently, this also mandates to include tsunami safe designs within the building codes for effective land use development. Further, these results also necessitate to educate the

coastal communities living along the coastline about tsunami risks and preparedness.

*Protective influence of natural features*

Conversely, the eastern regions of Greater Karachi, comprising mainly the District of Malir, benefit from the protective influence of creeks along the coastline (see Fig. 9(d) and Fig. 10(d)). These natural features act as a barrier, reducing the

vulnerability of shoreline areas to tsunami waves and the full impact of the inundation. Emergency management efforts should consider leveraging these natural defences to enhance resilience in these areas. Further, these creeks require protection from anthropogenic activities to sustain this region as a natural barrier including restoring and preserving mangroves and other coastal vegetation.





Other than flooding, early warning systems need to be tailored in the generalized zones to account for the wave
arrival times, ensuring that residents receive alerts with sufficient lead time for evacuation. In this context, a separate
approach needs to be considered for the two regions given the wave arrive early in the western region which comprises high
uplands and the more vulnerable central regions. Whereas the eastern regions shielded by creeks experiences delayed arrival
times.

**6.2 Late arrival of waves**

The late-arriving waves, such as those observed in Pasni, Karachi, and Seychelles from 1945 tsunami, warrant a yellow
caution flag about the arrival times. Figure 11 summarizes Table A1, which clearly shows wave arrivals were later for
observations compared to simulations with the difference ranging between 2 and 3.5 hours. The reason has been attributed to
delays possibly due to secondary sources such as submarine slides. Notably, Bombay did not experience any delays in wave
arrival compared to simulations, whereas Seychelles, despite being farther away, encountered delays. Therefore, emergency
managers should adopt a cautious approach, extending the watch duration and emphasizing the need for continued
monitoring and preparedness even after the initial threat assessment.

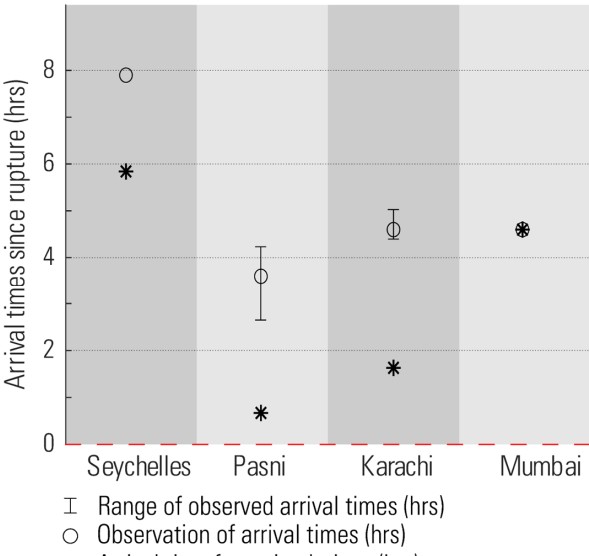

**Figure 11:** Paired Arrival Times of Tsunami Waves: A Comparison of Localities


**7 Implication for tsunami risk**

This study offers critical insights into the tsunami risk posed by hypothetical rupture scenarios originating from the Makran

Subduction Zone (MSZ) and their potential impact on the North West Indian Ocean coast, with a particular focus on the greater Karachi coastline. Here are the five key findings:

*Regional Variations:* Closer proximity to the earthquake source results in higher wave heights and shorter arrival times. For instance, Pasni experienced a peak wave height of 2.1 meters above mean sea level (MSL) with an arrival time of 32 minutes, while locations farther away, like Bombay, saw wave heights of 0.36 meters arriving 4 hours and 36 minutes

post-event.

*Influence of Rupture Area:* Larger rupture areas generate higher wave heights, with Scenario 800x355 producing waves up to 9.09 meters high at Muscat and 7.76 meters high at Gwadar. Conversely, narrower rupture areas lead to lower wave heights, such as in Scenario 220x210 with Gwadar experiencing 4.46 meters high waves.

*Impact on Karachi:* The width of the rupture area significantly influences wave heights along the Karachi coast. For

example, at Cape Montz, wave heights varied from 0.77 meters in Scenario 150x100 to 2.94 meters in Scenario 800x355, with corresponding arrival times ranging from 1.14 to 0.91 hours post-event.

*Role of Geographic Features:* Natural features like creeks and higher topography play crucial roles in mitigating tsunami impacts. Locations like Rehri Goth and Lath Basti benefit from creeks, experiencing reduced wave heights, while the western region of Karachi with elevated terrain experiences less severe inundation.

*Implications for Emergency Management:* Karachi faces unique challenges as a mid-field tsunami zone, with limited time for preparation and response. To address this, the study suggests dividing Karachi into two zones based on vulnerability, with Karachi Port facing higher immediate risks compared to Port Qasim.

These findings underscore the importance of comprehensive disaster management strategies tailored to Karachi's unique geographic and tsunami risk profile. With potential wave heights ranging from mere centimetres to several meters

and arrival times varying from minutes to hours, effective preparedness and early warning systems are crucial for mitigating the impact of tsunamis on the Karachi coastline.

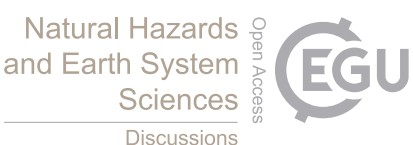
## Appendix A    Tables

**Table A1** Time of wave since 1945 earthquake; reported vs simulated for various cities

| Location | Pasni | | Karachi | | Bombay | | Seychelles |
|---|---|---|---|---|---|---|---|
| Evidence of largest wave | Eyewitness interviewed in 2014 During Fajr Prayer* (Kakar et al (2015), page 31) | Town flooded (Pendse, 1948) | Newspaper accounts (chiefly Times of India, 30/11/1945) | Tide gauge disabled (Atwater et al., 2020, page 19) | Newspaper account (Times of India, 29/11/1945) | Marigram (Neetu et al., 2011) | Observed water level at Port Victoria (Bear and Stagg, 1946) |
| Reported time of wave | During Fajr Prayer 28/11/1945 | 28/11/1945 07:15 IST | 28/11/1945 08:15 IST | Gap in marigram between 08:06 and 08:30 IST | 28/11/1945 08:15 IST | 28/11/1945 02:37 UTC | 28/11/1945 05:52 GMT |
| Corresponding time since 1945 earthquake (hrs) | 2.8 to 4.1 | 3.6 | 4.6 | 4.5 to 4.9 | 4.6 | 4.6 | 7.9 |
| Time of maximum water level, in hrs after earthquake in scenrio 150x100 | 0.67 | 0.67 | 1.63 | 1.63 | 4.6 | 4.6 | 5.84 |
| Time Difference in hrs between observation and simulation | 2.13 to 3.43 | 2.93 | 2.97 | 2.87 to 3.27 | 0 | 0 | 2.06 |

| Time of earthquake: | GMT: 27/11/1945 21:57 |
|---|---|
| | IST:   28/11/1945 03:37 |

*Today, the time window for Fajr prayer on 28th November in Pasni is 05:52 till 07:13 Pakistan Standard Time (PKT; attributed to University of Islamic Sciences, Karachi). The equivalent times in IST are 06:22 and 07:43.



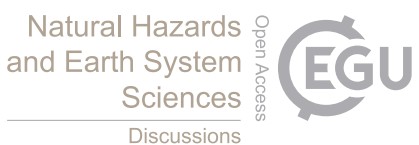
**Table A2** Summary of the literature modelling tsunami hazard along the shores of the Northwest Indian Ocean due to the Makran Subduction Zone.

| Study | Tsunami sources | Coast | DTHA/ PTHA | Authors main points |
|---|---|---|---|---|
| Heidarzadeh et al. (2007) | Mw 8 offshore Chabahar | Iran, Oman, Pakistan, and India | DTHA | First tsunami wave hits the nearest coastline within about 15 to 20 minutes. Wave hits all four countries within the region after about 1hr. Maximum wave heights reach to about 3 meters along the coast. |
| Rajendran et al. (2008) | 1945 eq. | Pasni, Muscat, Karachi, Mumbai, Goa, Karwar, Kerala | DTHA | Delay in arrival time at Pasni, Karachi and Mumbai formerly Bombay is attributed to a secondary submarine landslide source |
| Heidarzadeh et al. (2008a) | 1945 eq. | Iran, Oman, Pakistan, and India | DTHA | Comparison of the run-ups observed from the 1945 event with the modelling reveal the tsunami cannot be explained only by a tectonic source and it is proposed one of the following reasons:<br>a) submarine slide<br>b) large displacement on splay faults<br>c) report of large run-ups being incorrect |
| Heidarzadeh et al. (2008b) | 1945 eq. | Iran, Oman, Pakistan, UAE | DTHA | The 1945 scenario replicated historical observations, with wave heights of 4-5 m in Pasni and 1.5 m at Karachi, arriving approximately 120mins post-quake. Along the coast of the southern coast of Iran and Oman wave heights were under 1 m. Moved along the coast, six tsunami scenarios considered for the maximum regional earthquake with maximum tsunami wave heights of 4-9.6m along the southern Iran and Pakistan coasts. Directivity plays a crucial role, with energy traveling perpendicular to the fault segment. Significant hazards also along the coast of Oman. |
| Heidarzadeh et al. (2009a) | Worst Case Scenario eqs. of Mw 8.6 and 9.0 having rupture length of 500km and 900km respectively.<br><br>Mw: 8.3 (maximum regional earthquake) used to simulate six different scenarios with each scenario at a different location along the MSZ.<br><br>Splay fault branching from the main plate boundary for Mw 8.6 | Iran, Oman, Pakistan | DTHA | The two worst-case scenarios resulted in runup heights of 12-18 meters for the first scenario and 24-30 meters for the second scenario. Splay fault branching from the main plate boundary can increase the wave height and the local run-up by a factor of 2. |


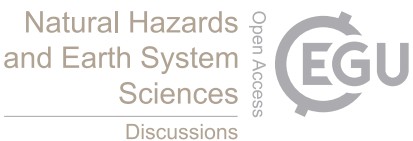

| Reference | Source / Scenario | Location | Method | Findings |
|---|---|---|---|---|
| Heidarzadeh et al. (2009b) | 1945 eq. used to simulate five different scenarios with each scenario at a different location along the MSZ. | Iran, Oman, Pakistan, UAE | DTHA | Significant tsunami hazard to the southern coasts of Iran and Pakistan, with waves reaching heights of 5-7 meters whereas the northern coast of Oman and eastern coast of the UAE face risks to a lesser extent. Urban areas like Karachi and Muscat experience significant wave amplification. Minor effects on Oman and UAE due to directivity. Emergency services have about 15 minutes to issue warnings. Maximum inundation distance from the 1945 Makran tsunami was about 1 km in Pasni. |
| Jaiswal et al. (2009) | 1945 eq. | Western Coast of India | DTHA | Tsunami waves originating from the Makran Subduction Zone in the Arabian Sea can have severe consequences in terms of inundation for the western coast of India, particularly Gujarat. Arrival times are found to be within half an hour and one hour along the coast of Gujrat. |
| Heidarzadeh & Kijko (2011) | $M_w$ 8.1 eq. source located at three tsunami-generating sub-regions along the Makran coast | Iran, Oman Pakistan | PTHA | A single moment magnitude was used to determine tsunami probability. Southern coasts of Iran and Pakistan including Muscat, Oman, are the were found to have a probability of having tsunami waves exceeding 5 meters over a 50-year period at 17.5%. For moderate tsunamis (between 1 and 2 meters) probability is 40% over a 50 year period. |
| Neetu et al. (2011) | 1945 eq. | Karachi, Mumbai, Port Victoria | DTHA | No need to invoke submarine landslides to explain late arrival of waves as late arrivals attributed to trapping of energy in the nearshore indicating presence of edge waves. |
| Heidarzadeh & Satake (2014a) | Proposed 1945 eq. alternative | Karachi, Mumbai, Port Victoria | DTHA | Tsunami source constrained through use of spectral analysis, tsunami forward modelling and tsunami inversion analysis. The earthquake source is determined to be a four-segment fault with varying slip amounts, including a slip of 4.3 m onshore near Ormara and a slip of 10 m offshore at a water depth of around 3,000 m. |
| Heidarzadeh & Satake (2014b) | Mud Volcano, submarine landslide, splay fault | Iran, Oman, Pakistan, India | DTHA | Tsunami backward ray tracing indicates that the probable source offshore of Jiwani, Pakistan, possibly triggered by the main earthquake. Modelling based on a mud volcano at the site newly |




| Reference | Scenario | Location | Method | Findings |
|---|---|---|---|---|
| Payande et al. (2015) | Four scenarios: 1945 eq, including three scenarios categorized as weak ($M_w$ 7.5), moderate ($M_w$ 8.1) and strong ($M_w$ 9.1) off shore Chabahar Bay. | Chabahar Bay (Iran) | DTHA | formed island in the Arabian Sea failed to replicate observed sea level data. Also ruled out mud volcanoes or diapirs as potential tsunami sources. A submarine slump with dimensions of approximately 10–15 km and a thickness of about 100 m, located 60-70 km off the coast of Jiwani, appears to accurately reproduce the observed sea level records. Splay faults were also ruled out as the numerical modelling did not align with the observed wave form. Kenarak coasts of Chabahar bay are identified as the most vulnerable to tsunamis due to their low elevation with respect to sea level. Tsunami wave amplification is observed in outer coasts of the bay, with waves reaching up to 18 meters in height. Two flood zones are identified: one with no or low inundation on cliffy shores, and another with extensive inundation in Kenarak areas, sandy beaches, and gentle slopes. Chabahar city parts are mostly located in low tsunami hazard zones, but in the case of strong earthquakes (above 9 Mw), all parts of Kenarak city would be in high tsunami hazard zones. |
| Rastgoftar & Soltanpour (2016) | 1945 eq. and submarine landslide | Pakistan, Iran | DTHA | The simulated landslide tsunami aligns well with reported wave heights at different coasts, and the calculated arrival times of tsunami waves at various regions of the Makran coasts match historical observations when the occurrence time of the submarine landslide is assumed to be about 3.5 hours after the earthquake. This is not possible with 1945 earthquake source. |
| Hoechner et al. (2016) | $M_w$ 7.4 – 9.4 | Pakistan, Iran, Oman | PTHA | Probability of exceedance shows strong dependence on maximum magnitude ($M_{max}$) over long hazard time than on short hazard times. For decision-makers Probabilistic tsunami height (PTH) rather than POE, as PTH is more linearly dependent on $M_{max}$, making it easier to manage the variations in the probability estimates. |
| El-Hussain et al. (2016) | $M_w$ 7.9 – 9.1 with interval of 0.2 | Entire Oman coast | PTHA | Probability of maximum wave amplitudes exceeding 1 meter reaches 0.7 to 0.85 for 100 and 250 year exposure times, respectively, in certain coastal locations. For longer exposure times (500 and 1000 years), the |




| Reference | Description | Location | Method | Findings |
|---|---|---|---|---|
| Heidarzadeh & Satake (2017) | Combined earthquake-landslide model for 1945 including splay faults | Pasni, Ormara, Karachi, Mumbai | DTHA | probability of exceeding 1 meter wave amplitude reaches 100% at some locations along the northern Omani coast. The northern coast, including Sur, Muscat, and areas facing the Sea of Oman, is identified as the most tsunami hazardous zone. The southern coast experiences less significant tsunami hazard, with most scenarios from the Makran subduction zone resulting in wave amplitudes below 1 meter. splay fault sources added to the existing earthquake source including delayed rupture scenarios, was not capable of reproducing the near-field runup height of 10-12 m observed during the 1945. 12 different landslide sources were considered but only on submarine landslide with 40 km³ volume located at 63.0° E and 24.8° N was found to be capable of reproducing the observation. |
| Hasan et al. (2017) | 1945 eq. proposed by Byrne et al. (1992) and Heidarzadeh and Satake (2017) | Karachi Port | DTHA | Modest tsunami wave heights can produce damaging currents, as demonstrated by the 1945 Makran tsunami at Karachi Port. Test against historical data, simulations show the tsunami generated ebb currents of 4 to 5 knots, leading to boat movement and groyne damage with use of damage index. The extension in the breakwater leads to weaker currents. |
| Rashidi et al. (2018a) | Full rupture divided into 20 segments with width of 210 km and a co-seismic slip of 10 m. | Iran, Oman, Pakistan, India | DTHA | Analysed the tsunami wave energy distribution from static and dynamic sea-floor deformation as a result of the distribution of seismic energy released. For every increase in magnitude by one unit, the associated tsunami wave energy becomes about 10^3 times greater. |
| Rashidi et al. (2018b) | Mw 8.7 eq. scenario in western Makran | Southeastern coastline of Iran, Oman | DTHA | Gulf of Oman traps tsunami waves, intensifying impacts on Iran and Oman. Virtual gauges placed along the southeastern Iranian coastline revealed 20 minutes for peak wave amplitudes, reaching up to 11 meters. Maximum tsunami amplitudes inside the Gulf of Oman and Arabian Sea Basins were 11 meters and 6 meters, respectively. Maximum computed run-up values were 10, 17, and 19 meters along the Makran coastline with corresponding |




| Reference | Scenario | Location | Method | Findings |
|---|---|---|---|---|
| Rashidi et al., (2018c) | $M_w$ 8.7 eq. scenario in western Makran | Southern coastline of Iran | DTHA | maximum inundation distances of 6, 6, and 4 km. Takes about 20 minutes for maximum tsunami wave amplitudes to be observed. The Gulf of Oman traps tsunami waves, leading to significant impacts on the coastlines of Iran and Oman. Maximum tsunami wave heights of 11m are observed along the southeastern coastline of Iran. Inside the Gulf of Oman and Arabian Sea Basins, maximum tsunami amplitudes reach up to 11m and 6m, respectively. Maximum run-up heights range from 10m to 19m, with corresponding inundation distances of 4km to 6km. |
| Sarker (2019) | Five earthquake scenarios with Mw 8.0 and 8.4 having 100 and 1000 year return period moved along the coastline including the case for a $M_w$ 7.8. | Duqm, Masirah, Muscat, Fujairah, Pasni, Karachi, Gujarat | DTHA | Location of fault rupture area west of the 1945 rupture area is found to be the worst. Worst affected areas are north of the Makran subduction zone |
| Rashidi and Keshavarz Farajkhah (2019) | Eastern Makran $M_w$ 7.5-8.9 Western Makran $M_w$ 7.5-8.8 Entire Makran $M_w$ 7.5-9.1 | Southeastern Iran coast | PTHA | Coastline of Konarak have the highest vulnerability and the lowest vulnerability is in Sirik with respect to probability of exceedance of wave height 3m in 500 years. Future attention needs to paid to long-term tsunami hazard assessment between Jask and Beris |
| Rashidi et al. (2020) | Based on Smith et al. (2013), 100 heterogenous slip distributions with mean coseismic slip of 10 m. Rupture spans entire MSZ with length 900 km and width 210 km | Iran, Pakistan and Iran | PTHA | Maximum tsunami wave heights reach 16m along the Iran and Pakistan coastlines with wave height variability influenced by slip distribution heterogeneity. Central Iran-Pakistan shoreline and the Muscat-Sur area exhibit considerable probabilistic tsunami hazards due to energy depletion along these regions. Iran shows higher vulnerability than Pakistan and Oman, with broader distribution of high tsunami wave heights along its shoreline. |
| Gopinathan et al. (2021) | Earthquake | Karachi Port | PTHA | statistical emulation, trained 300 tsunami simulations, used to predict 1 million large tsunamis originating from the Makran Subduction Zone. For range of magnitudes Mw 7.5-8.8, maximum velocities (extreme events) and wave heights reaching up to 16 ms$^{-1}$ and 8 meters are observed, respectively |
| Haider et al. (2023a) | integrated five | Karachi, | Multi- | Recurrence interval for mega-tsunamis (≥12 m) in the |

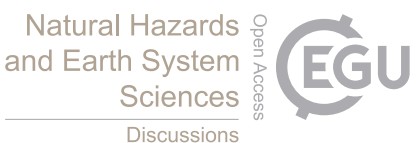
| Reference | Scenarios | Study Area | Method | Results |
|---|---|---|---|---|
| | different approaches to evaluate the recurrence intervals, sources, and potential impacts of tsunamis: 1. PTHA 2. DTHA 3. Geophysical-seismic (2-D thermal modeling) 4. Sedimentary tsunami deposits analysis 5. Historical record analysis additionally, cyclone risk is assessed by interpolating storm tracks dating back to 1842 CE. | Ornara, Pasni and Gwadar | proxy approach | Arabian Sea: 500-1000 years with seismic scenarios accounting for 60% of megatsunamis and 40% accounting for secondary landslide including others. hazard analysis helped to identify four wave scenarios (3, 7, 10, and 15 m), with results indicating that waves of 10 m and 15 m could severely disrupt the study area. frequency of cyclones has increased significantly in the past 64 years, with the intensity jumping from Tropical Storm (TS) to Category-3. |
| Haider et al. (2023b) | 1945 eq. for calibration | Gwadar, Pasni | DTHA | dynamic and static approaches for modeling tsunami inundation and risk analysis. three different wave scenarios - 7 m, 10 m, and 15 m are used high vulnerability of Gwadar and Pasni to tsunamis, particularly with wave heights ≥7m and lengths ≥15 km. demonstrate the potential devastating impact of a 15m wave scenario on both cities by determining the damage probability |
| Nouri et al. (2023) | 4 submarine landslide scenarios having volumes 10–40 km3 at 25 locations in western Makran 100 hypothetical scenarios. | Western Makran which includes Iran and Oman | DTHA | Oman's coastline is more vulnerable to the hazard from landslide-generated tsunami compared to Iran, with a maximum average wave height of 3.1 m compared to 0.9 m for Iran. Muscat, Oman, is more vulnerable to this type of tsunami compared to Chabahar, Iran due to the significant gap between the maximum and average wave height for all scenarios. |





**Table A3:** Parameters for earthquake scenarios

| Scenarios | Fault Length (km) | Fault Width (km) | Top center Coordinates | | Depth (km) | Slip (m) | Strike (º) | Rake (º) | updip edge (º) |
| --- | --- | --- | --- | --- | --- | --- | --- | --- | --- |
| | | | Long. (º E) | Lat. (º N) | | | | | |
| *Byrne et al. (1992)* | 150 | 100 | 63.9454 | 24.8797 | 20 | 7 | 270 | 89 | 7 |
| *Smith et al. (2013)* | 220 | 355 | 63.2084 | 22.987 | 20 | 10 | 270 | 89 | 7 |
| | 220 | 210 | 63.2084 | 23.6401 | 20 | 10 | 270 | 89 | 7 |
| | 400 | 355 | 64.0246 | 24.13 | 20 | 10 | 270 | 89 | 7 |
| | 400 | 210 | 64.0246 | 24.67 | 20 | 10 | 270 | 89 | 7 |
| | 800 | 355 | 62.124 | 24.13 | 20 | 10 | 270 | 89 | 7 |
| | 800 | 210 | 62.124 | 24.67 | 20 | 10 | 270 | 89 | 7 |





**Table A4:** Numerical values of Tsunami Arrival times (hrs), wave heights (m) and uplift/subsidence (m) at various cities of the North West Indian Ocean region including Port Victoria, Seychelles plotted in **Fig. 5(a)**

| | | Seychelles | Jask | Muscat | Chahbahar | Gwadar | Pasni | Ormara | Karachi | Mandvi | Mumbai |
|---|---|---|---|---|---|---|---|---|---|---|---|
| **Scenario (1945) 150x100** | Wave Height | 0.11 | 0.65 | 0.51 | 0.74 | 0.94 | 2.01 | 1.92 | 0.89 | 0.78 | 0.36 |
| | Arrival Time | 5.84 | 2.43 | 1.40 | 2.79 | 0.75 | 0.67 | 0.54 | 1.70 | 3.41 | 4.60 |
| | Uplift/subsidence | 0.0000 | -0.0025 | -0.0020 | -0.0098 | -0.0132 | 0.8519 | 0.8905 | -0.0079 | -0.0016 | 0.0000 |
| **Scenario 220x355** | Wave Height | 0.35 | 1.45 | 1.45 | 2.05 | 4.46 | 4.09 | 2.83 | 2.19 | 1.31 | 0.94 |
| | Arrival Time | 5.76 | 1.42 | 1.60 | 0.80 | 0.56 | 0.67 | 0.60 | 1.78 | 3.36 | 4.52 |
| | Uplift/subsidenc | 0.0000 | -0.0202 | -0.0113 | 0.0626 | 0.5862 | 1.2315 | 0.2536 | -0.0262 | -0.0085 | 0.0000 |
| **Scenario 220x210** | Wave Height | 0.40 | 1.21 | 1.20 | 1.87 | 4.28 | 4.04 | 2.50 | 1.62 | 1.24 | 0.86 |
| | Arrival Time | 6.67 | 1.47 | 1.21 | 2.10 | 0.23 | 0.57 | 0.54 | 1.79 | 3.44 | 4.62 |
| | Uplift/subsidence | 0.0000 | -0.0138 | -0.0084 | 0.0151 | 1.3374 | 1.3965 | 1.1320 | -0.0203 | -0.0055 | 0.0000 |
| **Scenario 400x355** | Wave Height | 0.55 | 1.49 | 1.77 | 2.09 | 4.54 | 4.50 | 5.85 | 3.79 | 1.88 | 1.20 |
| | Arrival Time | 5.80 | 1.41 | 1.61 | 0.61 | 0.56 | 0.79 | 0.50 | 1.44 | 3.19 | 4.51 |
| | Uplift/subsidence | 0.0000 | -0.0306 | -0.0192 | -0.0330 | 0.6948 | 1.2701 | 1.2838 | -0.0087 | -0.0146 | 0.0000 |
| **Scenario 400x210** | Wave Height | 0.51 | 1.25 | 1.23 | 1.52 | 5.08 | 4.11 | 5.29 | 2.33 | 1.92 | 1.19 |
| | Arrival Time | 5.88 | 1.47 | 1.21 | 0.75 | 0.67 | 0.57 | 0.39 | 0.77 | 3.32 | 4.62 |
| | Uplift/subsidence | 0.0000 | -0.0205 | -0.0134 | 0.0367 | 1.4038 | 1.4046 | 1.4641 | 0.0320 | -0.0094 | 0.0000 |
| **scenario 800x355** | Wave Height | 0.90 | 4.91 | 9.09 | 8.47 | 7.76 | 4.58 | 5.88 | 3.77 | 1.94 | 1.55 |
| | Arrival Time | 5.74 | 0.54 | 0.66 | 0.45 | 0.54 | 0.58 | 0.50 | 1.43 | 3.19 | 4.83 |
| | Uplift/subsidence | 0.0000 | 1.1645 | 0.1855 | 1.2105 | 1.2612 | 1.2222 | 1.2424 | -0.0214 | -0.0253 | 0.0000 |
| **Scenario 800x210** | Wave Height | 0.86 | 4.33 | 7.33 | 7.15 | 6.22 | 4.12 | 5.27 | 2.19 | 1.98 | 1.32 |
| | Arrival Time | 5.84 | 0.44 | 0.76 | 0.34 | 0.68 | 0.47 | 0.39 | 0.75 | 3.32 | 4.63 |
| | Uplift/subsidence | 0.0000 | 1.1700 | 0.0529 | 1.2818 | 1.6014 | 1.3433 | 1.4316 | 0.0437 | -0.0161 | 0.0000 |






**Table A5:** Numerical values of Tsunami Arrival times (hrs) and wave heights (m) at various locations along the coast of Karachi plotted in **Fig. 6**

| Location | Scenario 800x210 | | scenario 800x355 | | Scenario 400x210 | | Scenario 400x355 | | Scenario 220x210 | | Scenario 220x355 | | Scenario (1945) 150x100 | |
|---|---|---|---|---|---|---|---|---|---|---|---|---|---|---|
| | Arrival Time | Wave Height | Arrival Time | Wave Height | Arrival Time | Wave Height | Arrival Time | Wave Height | Arrival Time | Wave Height | Arrival Time | Wave Height | Arrival Time | Wave Height |
| Cape Monze (66.66,24.82) | 0.75 | 2.01 | 0.91 | 2.94 | 0.76 | 2.28 | 0.91 | 2.86 | 1.18 | 1.09 | 1.13 | 1.32 | 1.14 | 0.77 |
| Goth Yusuf Ali (66.72,24.83) | 0.86 | 1.98 | 1.02 | 3.09 | 0.87 | 2.24 | 1.02 | 2.96 | 1.27 | 1.09 | 1.46 | 1.49 | 1.22 | 0.78 |
| Goth Jumma (66.74,24.84) | 0.92 | 2.01 | 1.07 | 3.36 | 0.93 | 2.17 | 1.08 | 3.23 | 1.41 | 1.26 | 1.43 | 1.47 | 1.29 | 0.85 |
| Paradise Point (66.78,24.84) | 0.98 | 2.10 | 1.14 | 3.50 | 1.01 | 2.32 | 1.13 | 3.37 | 1.44 | 1.37 | 1.38 | 1.65 | 1.35 | 0.98 |
| French Beach (66.82,24.84) | 1.01 | 2.08 | 1.16 | 3.45 | 1.03 | 2.22 | 1.16 | 3.40 | 1.48 | 1.40 | 1.44 | 1.68 | 1.41 | 1 |
| Hawksbay (66.85,24.85) | 1.18 | 2.10 | 1.32 | 3.52 | 1.19 | 2.26 | 1.32 | 3.40 | 1.66 | 1.55 | 1.72 | 1.97 | 1.59 | 1.05 |
| Kakapir (66.90,24.83) | 1.21 | 2.22 | 1.34 | 3.98 | 1.24 | 2.40 | 1.34 | 3.91 | 1.71 | 1.70 | 1.72 | 2.23 | 1.62 | 1.08 |
| Sandspit (66.92,24.83) | 1.20 | 2.23 | 1.33 | 4.25 | 1.29 | 2.31 | 1.34 | 4.18 | 1.70 | 1.82 | 1.74 | 2.34 | 1.62 | 1.11 |
| Manora (66.97,24.79) | 1.23 | 2.21 | 1.37 | 4.20 | 1.25 | 2.48 | 1.37 | 4.13 | 1.77 | 1.62 | 1.78 | 2.39 | 1.63 | 0.94 |
| Dolmen Mall (67.02,24.79) | 1.52 | 2.04 | 1.61 | 3.42 | 1.60 | 2.11 | 1.62 | 3.32 | 2.06 | 1.84 | 2.04 | 2.39 | 1.93 | 1.09 |
| Seaview (67.05,24.77) | 1.44 | 2.29 | 1.54 | 4.19 | 1.46 | 2.38 | 1.55 | 4.19 | 2.00 | 2.15 | 1.97 | 2.87 | 1.92 | 1.26 |
| Crescent Bay (67.07,24.74) | 1.71 | 1.65 | 1.58 | 2.81 | 1.73 | 1.82 | 1.58 | 2.78 | 2.17 | 1.57 | 2.12 | 1.90 | 2.08 | 0.91 |
| Creek Park (67.10,24.78) | 2.02 | 1.22 | 1.88 | 1.66 | 2.06 | 1.28 | 1.93 | 1.61 | 2.39 | 1.06 | 2.33 | 1.26 | 2.33 | 0.73 |
| Marina club (67.08,24.80) | 2.09 | 1.40 | 1.96 | 1.85 | 2.13 | 1.48 | 1.98 | 1.79 | 2.55 | 1.25 | 2.52 | 1.45 | 2.45 | 0.83 |
| Ibrahim Hyderi (67.14,24.78) | 2.13 | 1.07 | 2.04 | 1.36 | 2.16 | 1.10 | 2.05 | 1.33 | 2.49 | 0.95 | 2.48 | 1.11 | 2.44 | 0.63 |
| Karachi fishiries (67.20,24.80) | 2.48 | 1.30 | 2.37 | 1.49 | 2.51 | 1.33 | 2.40 | 1.47 | 2.89 | 1.13 | 2.86 | 1.30 | 2.84 | 0.76 |
| Rehri Goth (67.24,24.81) | 2.81 | 0.79 | 2.52 | 0.86 | 2.75 | 0.80 | 3.02 | 0.88 | 3.06 | 0.66 | 3.02 | 0.75 | 3.02 | 0.48 |
| Lath Basti (67.26, 24.81) | 3.17 | 0.71 | 3.25 | 0.80 | 3.19 | 0.71 | 3.25 | 0.81 | 3.41 | 0.53 | 3.41 | 0.58 | 3.34 | 0.41 |
| Port Qasim (67.31,24.79) | 2.81 | 1.14 | 2.89 | 1.25 | 2.77 | 1.14 | 2.87 | 1.27 | 3.08 | 0.95 | 3.02 | 1.07 | 3.04 | 0.62 |
| Russian Beach (67.41,24.78) | 3.24 | 1.02 | 3.37 | 1.09 | 3.24 | 1.03 | 3.35 | 1.10 | 3.56 | 0.85 | 3.51 | 0.94 | 3.50 | 0.59 |





**Appendix B     Methodology on the development of the Digital Elevation Model for greater Karachi and it's nearshore**

A digital elevation model (DEM) was developed for Karachi that represents the latest topographic and bathymetric features. The DEM developed by Hasan et al. (2017) for their study on assessing tsunami currents was restricted to Karachi Port only. The current DEM includes recent developments which include the addition of the deep-water container terminal and its

approaches to Karachi Port (See Fig. 1 (c)). Furthermore, the DEM has been extended from the western most point of Karachi which includes Cape Monze to the eastern most point which includes Port Qasim and its approaches.

**Bathymetery**

For the development of the bathymetric part of the DEM, bathymetric soundings were extracted from Nautical Charts available from the Pakistan Navy (Bhatti, 2011, 2012, 2017a, b). The dredged channels approaching the Karachi Port and

Port Qasim are digitized from the Nautical Charts. For the off-shore regions where bathymetric data is not covered by Nautical Charts, the GEBCO data from 2020 with a global coverage of 15 arc-second is utilized (GEBCO Bathymetric Compilation Group 2020).

Tidal shorelines, which include both high water and low water lines, were extracted from (ESRI World Imagery, 2021) with reference to Nautical Charts. High water extents were identified through visual inspection of the (ESRI World

Imagery, 2021) which can also be verified on-site. Low water lines were extracted from Nautical charts keeping in view the variability of shorelines from the historical imagery (ESRI World Imagery, 2021).

**Vertical Datum**

The depths in nautical charts are referenced to the chart datums. For Karachi Port, the datum is 4.39 m below the new tidal benchmark at KPT reportedly referenced to the old one. The historical datum usable until 1992 across the channel at Manora

is 0.09 m below this new tidal benchmark. Further east at Port Qasim, the datum is 6.38 m below a benchmark located at the Port Qasim office building at Pipri (Pakistan Navy Hydrographic Department, 2018). Fig. 2 illustrates the vertical datums at the tide gauges for the two ports including the datum in 1945 at the old tide gauge location in Karachi.

The tsunami modelling domain covers approximately the entire 90 km shoreline of Karachi from Cape Monze to Port Qasim. The mean sea level (MSL) varies from approximately 1.8m at Karachi Port to 2.0 m as we move further east

towards Port Qasim. These variations in MSL are listed in tidal levels mentioned in PAK-20 (Bhatti, 2012). The entire bathymetric data is then shifted to 1.8 m which is the mean sea level at Karachi Port. Consequently, all depths, low water, high water lines, and dredged parts were transformed and shifted to MSL. All heights above MSL were considered positive and depths were considered negative.





**Topography**

The upland topography in the DEM is represented by gridded data of the Shuttle Radar Topography Mission (SRTM) having a spatial resolution of 1-arc second or 30m approximately (Nikolakopoulos et al., 2006b). The accuracy of DEM was also assessed in this study because hydraulic models, such as GeoClaw, could be sensitive to large variations in elevations. An inaccurate DEM may result in errors that may exceed the amplitude and wave height simulated for tsunami inundation analyses.

As part of the current study, the vertical accuracy of SRTM DEM was also assessed with reference to Ground Control Points (GCPs) acquired through the GNSS survey. The vertical accuracy for the study area, Karachi, was found within the global SRTM mission goal, under which Mean Absolute Error (MAE) and Root Mean Square Error (RMSE) stand at 16m and ±10m at a global scale for SRTM data (Farr et al., 2007). For Karachi, the MAE and RMSE values stood only at 3.161m and ±3.667m, respectively, lower than other similar studies in Asia (Mukul et al., 2015). In addition to this, a

positive correlation between SRTM and GNSS datasets is witnessed where the predictive pattern seemed to be moved in compliance with GNSS elevations.

However, the area 3km offset from the shoreline the data is replaced by a 1m high-resolution DEM model developed from stereoscopic orthoimages (Digital Globe, 2016). This area is illustrated in Fig. 1 (c). The elevations for both set of topographic data are referenced to globally modelled mean sea level (EGM 96). However, it is not known how

accurately EGM 96 values approximate the actual mean sea levels in the Karachi region. However, it has been assumed as to be closer to the actual mean sea levels.

**Final Digital Elevation Model**

To construct the final DEM we used a similar methodology to Hasan et al. (2018) whereby the topographic and bathymetric regions are interpolated separately in Surfer using the Minimum Curvature method, which employs spline interpolation

algorithms (Smith & Wessel, 1990). To avoid artefacts in the interpolated surface, the bathymetric data was interpolated in zones having varied spatial resolution of point depths. The zones are then mosaiced and resampled at 10m resolution with nearest neighbour resampling technique. The interpolated gridded surface is then validated with actual contours from referenced Nautical Charts. The final DEM has a 10m resolution and is referenced to a coordinate system of the World Geodetic System of 1984 (WGS84). This DEM offers a comprehensive representation of Karachi's topography and

bathymetry, crucial for various applications including tsunami modelling.



**Appendix C    Figures**

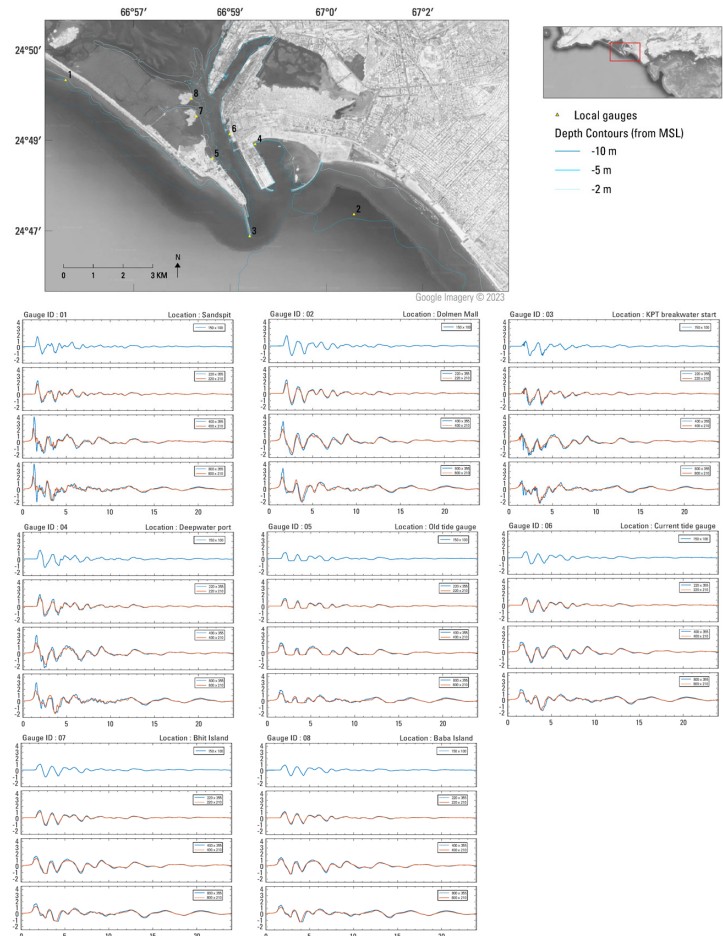

**Figure C1:** Time series for the water surface elevation at various synthetic tide gauges in and around Karachi Port



Natural Hazards
and Earth System
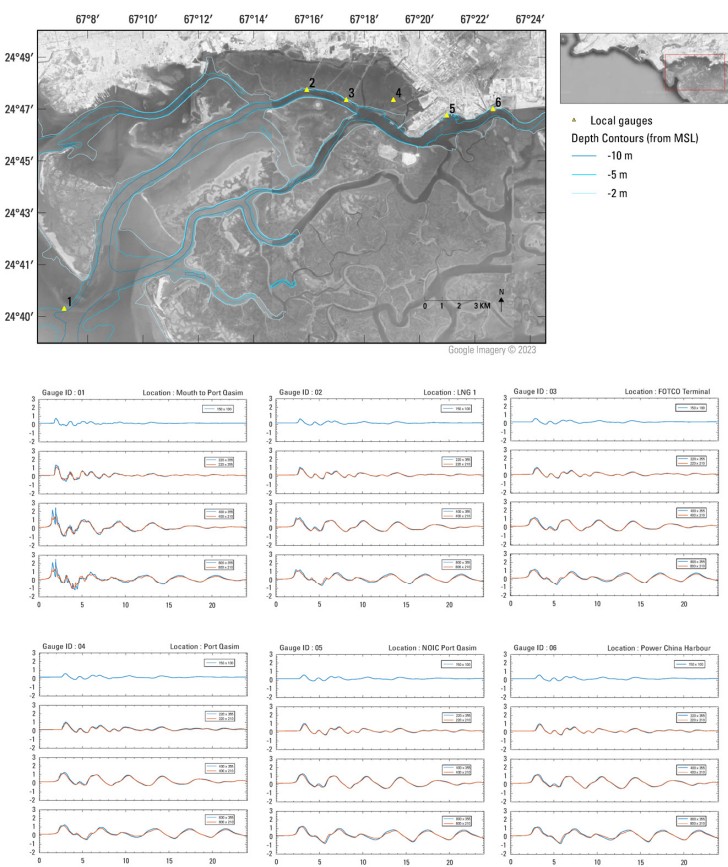

**Figure C2:** Time series for the water surface elevation at various synthetic tide gauges in and around Port Qasim



**Competing interests**

The authors declare no competing interests.

**Funding**

This work was initiated under a UNDP project supported by the United Nations Development Programme.

**Authors' contributions**

HH: conceptualization, formal analysis, investigation, methodology, visualization, writing (original draft), supervision. HAL: conceptualization, investigation, methodology. SA: investigation, formal analysis. SK: investigation, formal analysis, visualization, writing (review and editing). AR: conceptualization, methodology. MMR: methodology, supervision

**Acknowledgements**

The project was carried out with the support of Exascale Open Data Analytics Lab, National Center for Big Data and Cloud
Computing (NCBC), NEDUET, under the domain of Tsunami Modelling. Additionally, we would like to acknowledge the support extended by Brian F. Atwater. His critical reviews, particularly in enhancing the clarity of our figures, have significantly helped in improving the manuscript.

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
