# Peer review of "Mid-field tsunami hazards in greater Karachi from seven hypothetical ruptures of the Makran subduction thrust"

_Natural Hazards and Earth System Sciences, 2024_

## Author Comment (AC1)

Reply in response to Journal reviewer 1 comments of

**"Mid-field tsunami hazards in greater Karachi from seven hypothetical ruptures of the Makran subduction thrust"**

a manuscript by Haider Hasan, Hira Ashfaq Lodhi, Shoaib Ahmed, Shahrukh Khan, Adnan Rais, and Muhammad Masood Rafi submitted for publication in *Natural Hazards and Earth System Sciences* (nhess-2024-110)

Revised manuscript title

"Mid-field tsunami hazards in greater Karachi from hypothetical ruptures of the Makran subduction thrust"

1. ***Discussion of Results:*** *The paper lacks sufficient discussion regarding the results, particularly in comparing the actual Karachi marigram with the simulated results. Other studies have presented more robust simulations between the real data from the 1945 tsunami and their results. The authors should indicate that the discrepancies between their findings and the actual 1945 tsunami waveform may come from their simplistic representation of the tsunami source. The differences observed between the simulations and the two marigrams may not solely be attributed to tide gauge errors, but also to our limited understanding of the 1945 tsunami source.*

   This study concentrates on the implications that the emergency managers might have to consider while planning for mid-field cities such as Karachi. The purpose of the paper is not rather to delve into the 1945 tsunami source. Though the authors agree that the discrepancies may be due to simplistic source and not solely due to the tide gauge error. The same has also been reflected even in the abstract. In response to your comment, we will include a brief discussion in the revised manuscript to clarify that while the discrepancies observed may be partly attributed to the limitations in the representation of the tsunami source, they also highlight the challenges associated with modelling historical tsunami events.

2. ***Title Adjustment:*** *I recommend removing the word "seven" from the title.*

   I agree that removing the word "seven" would make the title more concise and focused. I will revise the title accordingly.

3. ***Abstract Corrections:*** *In Abstract (line 13), "km" should be corrected to "km²." Additionally, in line 21, the repeated use of "neither" makes the text unnecessarily verbose.*

       i. I have considered your comment regarding the use of "km" in line 13. The dimensions "$100 \times 150$ km to $355 \times 800$ km" are intended to specify the length and width of the rupture area. To ensure clarity I will include the units of both dimensions: $100$ km $\times 150$ km to $355$ km $\times 800$ km. Therefore, I will retain "km" to accurately represent the dimensions of the rupture areas.

       ii. I will revise line 21 to avoid the repeated use of "neither," making the text more concise.

4. ***Typos and References:*** *The manuscript contains several typographical errors, such as "Tohuku" (which should be "Tohoku") and "20011" (which should be "2011"). Moreover, in several instances,*

*the authors have neglected to include the publication year of references in parentheses. A thorough review of the text for typos and potential grammatical errors is recommended.*

Thank you for your careful review and for pointing out the typographical errors and inconsistencies in the references.

    i.    I will correct the typographical errors, such as changing "Tohuku" to "Tohoku" and "20011" to "2011."

    ii.    I will thoroughly review the manuscript to identify and correct any other typographical or grammatical errors.

    iii.    I will ensure that the publication years for all references are included in parentheses where appropriate.

5. ***Figure Combination:*** *It appears that Figures a&b could be combined into a single figure unless additional details (e.g., an earthquake catalog) are added to panel (a).*

I presume this comment refers to Figure 1 (a) and 1 (b). After careful consideration, I believe it is important to retain these figures Figures 1 a and b separately to maintain clarity and avoid clutter.

Figure 1 (a): Geographical Overview: Arabian Sea and surrounding nations. This figure provides a broad geographical context, which is crucial for understanding the regional setting of the study area.

Figure 1 (b): Dimensions and locations of hypothetical rupture scenarios: Makran Subduction Zone. This figure focuses specifically on the detailed dimensions and locations of the hypothetical rupture scenarios, which is essential for comprehending the specifics of the simulations.

Combining these figures into a single panel would make it cluttered and potentially obscure important details. Therefore, to ensure clarity and effective communication of the information, I would like to retain Figure 1 (a) and Figure 1 (b) as separate entities.

6. ***Model Details:*** *Additional information about the GeoClaw model should be included, such as inputs and outputs, the types of water equations used, the algorithms employed for inundation calculations, the nature of the friction effects (whether constant or variable), and whether structured or unstructured grids are used.*

Given that GeoClaw is a well-established model in the field, we aimed to focus on the aspects most pertinent to our study. However, to address your query more comprehensively:

    i.    **Model Inputs and Outputs:** GeoClaw utilizes a range of inputs including topography and bathymetry data, initial sea levels, and earthquake parameters. The model outputs are primarily related to water elevations, inundation extents, and flow velocities.

    ii.    **Water Equations Used:** GeoClaw is based on the shallow water equations, which account for conservation of mass and momentum. These equations are well-suited for modeling tsunami dynamics and inundation processes.

iii.  **Algorithms for Inundation Calculations:** The model employs a high-resolution shock-capturing finite volume method, which is enhanced by adaptive mesh refinement (AMR). This approach allows for detailed resolution of tsunami waves and inundation areas while efficiently managing computational resources.

iv.  **Friction Effects:** GeoClaw incorporates bed friction effects through the Manning coefficient. This parameter can be adjusted based on the varying characteristics of the coastal and bathymetric surfaces within the model domain.

v.  **Grid Structure:** GeoClaw uses structured rectangular grids that adapt dynamically through AMR. This higher resolution in areas of interest, such as nearshore regions, while maintaining computational efficiency.

We hope this additional information provides a clearer understanding of the model setup and its application in our tsunami hazard simulation for Karachi.

7. *Font Size in Figures: The font size in Figure 2 should be increased for better visibility.*

I will ensure the text within figure is easily readable.

8. *Scenario Simplifications: More details are needed regarding the simplifications made in selecting scenarios. Such simplifications (e.g., rectangular sources and uniform slip) could lead to potentially underestimated or less complex wave height distributions. The strike angle of 270 degrees appears inconsistent with the tectonics of the Makran thrust; any alterations in the strike direction could yield different wave directions and results. These simplifications require a more thorough explanation. Furthermore, these scenarios do not seem to offer new insights compared to those proposed in previous studies.*

We acknowledge the simplifications made in our scenario selection and their potential implications. Several simplifications were made in selecting our scenarios to standardize the analysis and align with previous studies, such as Smith et al. (2013). First, we chose to represent rupture areas as rectangles. While real earthquake ruptures are often more complex in shape, using rectangular sources simplifies the modeling process and provides a clear basis for comparison across different scenarios.

We also assumed a uniform slip distribution across the rupture areas. In reality, slip can vary significantly across the fault plane, which may lead to more complex and varied wave height distributions. This simplification could result in potentially underestimated wave heights, especially in regions where slip is concentrated.

The strike angle was assumed to be 270 degrees, which may not fully align with the tectonic structure of the Makran thrust. Variations in the strike direction could produce different wave directions and characteristics, affecting the results of the tsunami hazard assessment.

While these simplifications help to standardize the scenarios and facilitate the analysis, they also introduce certain limitations. Recognizing these limitations is crucial for understanding the scope and applicability of the results. Further refinement of the models, including more complex slip

distributions, non-rectangular rupture geometries, and different strike angles, could provide a more detailed understanding of the potential tsunami hazards in the Makran subduction zone.

We will revise section 3.2 to clarify the reasons behind the simplifications and acknowledge the implications, so that the section aligns with the request for more thorough explanations.

9. ***Table A3 Enhancements:*** *In Table A3, please add an additional column indicating the moment magnitude (Mw) for each scenario.*

I agree that this information is crucial for a comprehensive understanding of each scenario, hence, will add a column to Table A3 to include the moment magnitude (Mw) for each scenario.

---

## Author Comment (AC2)

Reply in response to Journal reviewer 2 comments of

**"Mid-field tsunami hazards in greater Karachi from seven hypothetical ruptures of the Makran subduction thrust"**

a manuscript by Haider Hasan, Hira Ashfaq Lodhi, Shoaib Ahmed, Shahrukh Khan, Adnan Rais, and Muhammad Masood Rafi submitted for publication in *Natural Hazards and Earth System Sciences* (nhess-2024-110)

Revised manuscript title

"Mid-field tsunami hazards in greater Karachi from hypothetical ruptures of the Makran subduction thrust"

We appreciate the feedback Reviewer 2 has provided on our manuscript. The review has highlighted several important areas for improvement. However, we would like to bring to your attention that Reviewer 1 accepted the paper for publication with minor revisions, citing its well-written nature and robust methodology as key strengths and the contributions of our study to the field. Reviewer 1's comments focused on specific enhancements, which we are already addressing.

Moreover, we understand Reviewer 2's concern about the length of the manuscript, it is primarily due to the inclusion of extensive figures and appendices that are integral to the study. This length has not been an issue for Reviewer 1 and as such, substantial reductions are not feasible without compromising the completeness of the results presented.

In contrast, some of the concerns raised in Reviewer 2's response appears to be in direct contradiction with the feedback from Reviewer 1. For example, while Reviewer 1 found our approach and results to be well-presented and methodologically sound, Reviewer 2 suggests that there are significant issues with our concept and methodology. It is important to note that many of these concerns relate to the presentation and clarification of our method rather than inherent flaws in the approach. To ensure that we address all concerns comprehensively, we have provided a point-by-point response to your comments and concerns.

1. *The article analyzes mid-field tsunami hazards. However, the concept of "mid-field" is presented rather vaguely. This concept is relative to "far-field" and "near-field", but the authors lack a clear definition in terms of distance. Moreover, they do not explain the significance of using this concept for disaster prevention. Compared to near-field events, are there any specific characteristics of mid-field tsunami propagation that need attention? How would it differ in causing inundation?*

   The introduction of the "mid-field" concept within this paper has not been used in tsunami hazard studies before, thus we recognize that its definition requires further clarification. Below we address the points raised which will help to improve upon this concept in the revised Introduction section of the manuscript.

**Clear Definition of Mid-Field**

The mid-field zone refers to coastal cities that are neither in the near-field (typically defined as locations where shaking from the tsunami-generating earthquake is strongly felt, usually within 100-200 km of the rupture and/or wave arrivals are within one hour) nor in the far-field (where the arrival of tsunami waves is delayed by three hours or more, often with the rupture around thousand or more kilometers away) (IOC, 2019; Wood and Council, 2011). Mid-field cities lie in an intermediate zone, where the tsunami is expected to reach in between 1 to 3 hours. While they are close enough to experience significant tsunami hazards, they are often too far from the rupture to feel strong seismic shaking. This absence of shaking can lead to a dangerous underestimation of the tsunami threat, as local populations may not receive the natural warnings that near-field communities experience. This also means that the population in the midfiled has to rely on timely dissemination of tsunami early warning by the authorities.

**Significance for Disaster Prevention**

The mid-field concept is critical for disaster preparedness because these cities, including places like Karachi, have limited lead times (often 1 to 3 hours) before the tsunami waves arrive. While far-field cities benefit from extended warning periods due to the longer travel time of tsunami waves, and near-field cities can rely on earthquake shaking as a natural alarm, mid-field cities may neither feel significant seismic activity nor have sufficient time for effective evacuation or disaster response if early warnings are delayed. Therefore, the introduction of mid-field is significant for improving hazard assessments, preparedness, and response strategies tailored to such locations. The paper underscores that current tsunami warning systems must be adapted to consider mid-field locations more rigorously to avoid underestimating the risks faced by these cities.

**Characteristics of Mid-Field Tsunami Propagation and Inundation**

In terms of tsunami propagation, mid-field cities typically experience the first waves within 1 to 3 hours of the rupture event, depending on their proximity to the source. These waves can often be higher and faster than those reaching far-field cities, but arrive without the natural warning signals that near-field cities rely on. As a result, mid-field cities need special attention in terms of evacuation planning.

For example, the simulations in our study show that for Karachi, the tsunami waves from a large rupture in the Makran Subduction Zone arrive at Karachi Port in 1.5 hours, whereas Port Qasim (~30 km east) experiences waves almost 3 hours later. This indicates that mid-field cities experience significant variability in arrival times and wave heights depending on local topography and coastal geography. Inundation in mid-field cities may differ from near-field events, primarily due to longer wave travel times, which may allow the tsunami to disperse somewhat, but the inundation in low-lying areas can still be severe, as shown in our simulations of Karachi. For

example, flooding can extend farther inland into residential zones without seismic shaking to prompt early evacuation.

2. *The selection of the seven cases in the article appears somewhat arbitrary. This study considers a maximum magnitude of up to 9.2, but it lacks a discussion on the probability of such an earthquake occurring. The author should at least clarify the magnitude represented by each case (Figure 3). If the intention is to select randomly, I suggest the author consider using the PTHA method, rather than focusing on the worst-case scenario.*

In response to this comment, we have provided a clarification below, which will be used to improve upon the section related Numerical model setup.

The approach used in our paper is based on Deterministic Tsunami Hazard Assessment (DTHA), with the focus on the largest possible events rather than estimating their probabilities. This method is useful in regions where historical data is insufficient to model event probabilities accurately, such as the Makran Subduction Zone (MSZ), where the recurrence intervals for large tsunamigenic earthquakes remain poorly understood. DTHA remains highly relevant for emergency planning, as it helps authorities plan for the maximum possible risk. This approach is similarly applied in other countries, including the United States (Dolcimascolo et al., 2021; Garrison-Laney et al., 2021) and Indonesia (Adityawan et al., 2023),where worst-case scenario modelling plays a critical role in hazard mapping and disaster management.

In line with the Sindh Tsunami Management and Response Plan (Provincial Disaster Management Authority (PDMA) Sindh, Not dated), our study emphasizes a worst-case scenario analysis, aligning with the approaches utilized by national authorities like the study by Pakistan Meteorological Department (PMD) discussed in the report by Mahmood et al., (2012). The Plan underlines the significant risks posed by a potential tsunami from the Makran Subduction Zone (MSZ) to Pakistan's coastal regions, particularly in densely populated areas such as Sindh, where the tsunami waves could reach the coast within a short timeframe. Given the substantial development and population growth since the last major tsunami in 1945, focusing on a worst-case scenario ensures that disaster preparedness, response planning, and mitigation efforts are sufficiently robust and tailored to the most severe potential impacts. Notably, both the Pakistan Meteorological Department (PMD) and the Provincial Disaster Management Authority (PDMA) utilize the deterministic approach, like our study, to inform tsunami risk assessments and enhance early warning systems.

In this context, the work by Adityawan et al. on the Palu region of Indonesia also consider the application of DTHA, particularly in areas where there is insufficient seismic data to model recurrence probabilities. Their research, which focuses on developing a tsunami early warning system based on maritime wireless communication, used numerical modelling of worst-case earthquake scenarios to enhance disaster preparedness. Like our approach, Adityawan et al.

emphasized the importance of focusing on worst-case scenarios in order to optimize early warning systems and improve mitigation strategies. Therefore, while PTHA offers important insights, our use of DTHA aligns with both international practices and the Sindh Tsunami Management and Response Plan.

Further, it is crucial to clarify that the selected scenarios are not arbitrary. They are based on a thermal modelling study conducted by Smith et al. (2013), which indicate the potential for megathrust earthquakes in this region. The scenarios reflect credible worst-case conditions derived from known tectonic characteristics and documented events. Specifically, the maximum magnitude of 9.2 is informed by Smith et al.'s thermal modeling, which suggests the possibility of a full-length rupture of the MSZ.

As suggested, we will annotate Figure 3 to specify the magnitude associated with each case and additionally, we will add a column to Table A3 to include the moment magnitude (Mw) for each scenario.

3. *There is some lack of clarity in the method description, such as the absence of a clear definition for arrival time. I also hope the author can explain why the presence of splay faults would result in the run-up doubling (Line 155). What is the reason behind this?*

It appears that the concerns raised here are more about specific terminologies and details in the literature review rather than the overall method description.

Firstly, regarding "arrival time," this term refers to the time it takes for the maximum wave to reach the coast after the tsunami is generated, as determined by our numerical simulations (line 283). We will update the text so the definition of "arrival time" is clearer at the outset in the manuscript.

As for the statement about the doubling of run-up heights due to splay faults, this is a finding from the study by Heidarzadeh et al. (2009a), cited in the literature review (Line 155). Splay faults are secondary fault structures that branch off the main fault and cause additional vertical displacement during an earthquake. This vertical movement increases the tsunami's energy and can lead to a doubling of the run-up heights as observed in the study. It should be noted the splay faults are not considered within our study, however, we will revise the manuscript to make this conclusion from Heidarzadeh et al. clearer.

4. *Figure 7 compares the simulated values with the records of the 1945 tsunami, yet the whole figure is perplexing. The negative phase of the blue curve (simulation) shows a significant truncation, which is caused by the wet point depth of the bathymetry and can introduce substantial errors.*

The negative phase of the blue curve (simulation) reflects the drawdown to the level of the modern tidal flat. This is clearly indicated in the figure. GeoClaw, the software employed in this

study, is adept at modelling wetting and drying processes thus ensuring accurate simulation of such effects. Given GeoClaw's capability to effectively handle such interactions, we believe that this aspect of the simulation will not introduce substantial errors.

5. *There are spelling issues throughout the entire article, with quite a few typos and grammatical errors. I suggest that the author thoroughly proofread and polish the text before resubmitting.*

We will thoroughly review the manuscript to address all spelling, typographical, and grammatical errors.

6. *Line 32: "December 20011"?*

Will make the correction

7. *Line 52: What do you mean "unseen"?*

By "unseen," we mean events that were not anticipated or previously experienced at those locations. Both the 2004 Sumatra-Andaman earthquake and the 2011 Tohoku earthquake occurred in regions that had not been historically associated with such large, destructive events.

8. *Line 102: What is the meaning of "dwarf"?*

The term "dwarf" here means the proposed fault rupture widths of 210-355 km are much larger than the 1945 rupture width of 100-150 km, making it seem smaller in comparison.

9. *Line 181-185: This section seems to deviate from the main topic; I suggest deleting it.*

The issues with the tide gauge during the 1945 tsunami, such as mechanical failures and sediment blockage, are crucial for understanding limitations in historical tsunami data. Including these details in the "Tsunami Modelling" section highlights challenges in interpreting historical records and underscores why some data may not align with model predictions. This context is essential for appreciating the complexities of tsunami modelling and justifies the need for accurate data collection and modelling improvements.

**References**

Adityawan, M. B., Nurendyastuti, A. K., Purnama, M. R., Arifianto, M. S., Farid, M., and Kuntoro, A. A.: Development of a tsunami early warning system on the coast of Palu based on maritime wireless communication, Progress in Disaster Science, 19, 100290, 2023.

Dolcimascolo, A., Eungard, D., Allen, C., LeVeque, R., Adams, L., Arcas, D., Titov, V., González, F., Moore, C., and Garrison-Laney, C.: Tsunami Hazard Maps of the Puget Sound and Adjacent Waters—Model Results from an Extended L1 Mw 9.0 Cascadia Subduction Zone Megathrust Earthquake Scenario: Washington Geological Survey Map Series 2021-01, 2021.

Garrison-Laney, C., LeVeque, R. J., and Adams, L. M.: Maritime Tsunami Hazard Assessment for the Port of Bellingham, Washington - Technical Report, 2021.

Heidarzadeh, M., Pirooz, M. D., and Zaker, N. H.: Modeling the near-field effects of the worst-case tsunami in the Makran subduction zone, Ocean Engineering, 36, 368–376, https://doi.org/10.1016/j.oceaneng.2009.01.004, 2009.

IOC: Tsunami glossary. Fourth edition., UNESCO, 2019.

Mahmood, N., Khan, K., Rafi, Z., and Løvholt, F.: Mapping of tsunami hazard along Makran coast of Pakistan, 2012.

Provincial Disaster Management Authority (PDMA) Sindh: Tsunami Management and Response Plan, Draft., Not dated.

Wood, N. and Council, N.: Tsunami Warning and Preparedness: An Assessment of the U.S. Tsunami Program and the Nation's Preparedness Efforts, 2011.

---

## Author Comment (AC3)

Reply in response to Journal reviewer #3 comments of

**"Mid-field tsunami hazards in greater Karachi from seven hypothetical ruptures of the Makran subduction thrust"**

a manuscript by Haider Hasan, Hira Ashfaq Lodhi, Shoaib Ahmed, Shahrukh Khan, Adnan Rais, and Muhammad Masood Rafi submitted for publication in Natural Hazards and Earth System Sciences (nhess-2024-110)

**SUMMARY**

**Mid-field**—Both reviewers #2 and #3 question our Karachi distinction between a tsunami near-field and a tsunami mid-field. The revised manuscript will address this shared concern by relating the mid-field concept to tsunami-safety challenges in greater Karachi, where time lags in communicating official warnings may exceed tsunami travel times.

**Overall strategy**—Reviewer #3 envisions modelling that is neither required by, nor suited for, today's practical purposes in greater Karachi. Official tsunami-inundation maps worldwide commonly rely on a deterministic modelling of a small number of scenario earthquakes. Our manuscript applies this same approach to improve official tsunami-inundation maps for greater Karachi. Those maps will meet an immediate need for guidance on plans for evacuation and land use. The maps will provide simplicity and transparency. A next generation of maps based on probabilistic assessment can await mature appraisal of the nearly endless geophysical possibilities that the Makran subduction zone offers.

In the detailed responses below, we address these comments and explain our revisions and the rationale behind them.

**1. MID-FIELD**

Reviewer #3 reports that our use of the mid-field concept is unsupported by cited precedents and, moreover, is inconsistent with tsunami-warnings in Pakistan. Those warnings, our manuscript noted, are intended to reach people at risk within the first quarter-hour after an earthquake. Reviewer #3 accordingly sees no point in placing Karachi in an intermediate position between a near-field where the parent earthquake provides an immediate tsunami warning, and a far-field where the earthquake would not be felt, and response would then depend on instrumental warning systems. We gratefully acknowledge these concerns in planning three revisions:

*Further definition of "mid-field" in a Karachi context*—With respect to tsunamis from the Makran subduction zone, greater Karachi occupies an intermediate position. It may receive no immediate warning from the seismic shaking felt in the near-field, while also lacking the long lead times for evacuation in the far field. The reviewed manuscript already makes this distinction, but the revised manuscript will state it more clearly.

*Comparisons with previous uses of "mid-field"*—The examples we know of refer to the 2004 Indian Ocean tsunami. Its mid-field shores included Sri Lanka (Fritz and Okal, 2008), along with Thailand and

parts of Indonesia remote from Aceh (Borrero et al., 2015). The tsunami arrival times in these areas were about two hours, similar to those we model for Makran tsunamis at Karachi. The losses of life in Sri Lanka and Thailand underscore the very risk we emphasize by applying "mid-field" to Karachi.

*Limitations of instrumental warning systems*—Reviewer #3 quotes the reviewed manuscript on the theoretical capability of tsunami early warning in Pakistan. But the reviewed manuscript went on to note practical limitations: "challenges in communication and infrastructure can cause official declarations to arrive too late, complicating timely disaster response (Witze, 2014), including for mid-field cities where the warning systems may not be fully effective (see sec 2.3)." In response to Reviewer #3, the revised manuscript will emphasize that although Pakistan's tsunami early warning system is designed to issue a public alert within 7-12 minutes, Karachi's mid-field position and current infrastructure gaps would limit the practical value of that alert. Revisions will note that greater Karachi has but three sirens with ranges up to 1.5 km (INP, 2021; UNDP, 2021)—two east of Karachi Port and one west of Karachi Port. Figure 1 will be amended to locate these sirens. The map will show that most are farther than 1.5 km from most of Karachi's vast coastline. That situation limits officials' ability to disseminate timely warnings to densely populated, low-lying areas. The revised manuscript will note, in addition, that effective warnings will require tsunami awareness programs tied to the very maps that the manuscript presents.

**2. OVERALL STRATEGY**

For Reviewer #3, the reviewed manuscript has a flaw more fatal than its use of "mid-field." That fatal flaw, the reviewer states, is its reliance on deterministic modelling of a small number of tsunami scenarios, in all of which slip is uniformly distributed on the Makran subduction thrust. These simplifications, the reviewer goes on to note, contrast with recent advances in relating Makran tsunami hazards to geodetic evidence, splay faults, stochastic approaches, and logic trees. The revised manuscript will address these concerns by showing Karachi's immediate need for a first generation of tsunami-inundation maps based on mature assumptions that users can understand.

*Precedents for deterministic models with uniform slip*—The revised manuscript will identify current examples of official tsunami-inundation maps that rely on uniform-slip scenarios:

- Chile's official tsunami inundation charts (CITSU), developed by Servicio Hidrográfico y Oceanográfico de la Armada (SHOA), represent maximum tsunami inundation under worst-case scenarios (http://www.shoa.cl/php/citsu.php). These SHOA maps, 73 in all, span nearly all of Chile's Pacific coast and range in publication date from 2006 to 2024.

- In the northwestern United States, current tsunami inundation and evacuation maps in Washington and Oregon represent near-worst-case scenarios from the Cascadia subduction zone. The main scenario used presupposes uniform slip on the subduction thrust, supplemented by slip on a splay fault. Washington State serves tsunami inundation maps "that are based on model results from hypothetical earthquake scenarios," each of which is identified in the map title (https://www.dnr.wa.gov/programs-and-services/geology/geologic-

[hazards/geologic-hazard-maps#tsunami-inundation](hazards/geologic-hazard-maps#tsunami-inundation)). These maps underpin evacuation maps that present a single inundation limit intended to approximate a worst case. Similarly in Oregon, that state's geology agency serves scenario-based inundation maps at [https://www.oregon.gov/dogami/pubs/pages/tim/p-tim-overview.aspx](https://www.oregon.gov/dogami/pubs/pages/tim/p-tim-overview.aspx) and derived evacuation maps at [https://www.oregon.gov/dogami/tsuclearinghouse/pages/tsunami-evacuation-maps.aspx](https://www.oregon.gov/dogami/tsuclearinghouse/pages/tsunami-evacuation-maps.aspx).

- Oman has the first official tsunami hazard maps for the Makran subduction zone. These were developed in 2014 by University of Cantabria, Spain (ESCAP, 2017). The modelers used the deterministic approach based on seven earthquake scenarios—the same number of scenarios that we have used in the reviewed manuscript. The largest earthquake considered was Mw 9.25.  ESCAP states, "Hazard maps showing inundation length and run-up heights have been developed on national scale (270 m resolution) and local scale (45 m resolution for 9 major cities) based on the analysis of tsunami propagation of 7 scenarios in the Makran region which have been considered as "worst credible" (with a maximum of Mw 9.25). In the subsequent discussions within the scientific community and with the Oman authorities, the probability of a Mw 9.25 event was questioned and considered very low. On this background, PACA decided to develop new scenarios based on Mw 8.25, which are still under development. The new scenarios will lead to smaller inundations zones and will certainly affect evacuation strategies and policies, which still need to be developed."

*Applicability to tsunami response and emergency management in greater Karachi*—These precedents do not erase the inherent limitations assessing tsunami hazards by means of a small number of scenarios with uniformly distributed slip. The revised manuscript will refer to Carvajal and Gubler (2016), on how uniform slip scenarios underestimate tsunami amplitudes compared to heterogeneous models; and to Geist and Dmowska (1999), on how uniform slip sources tend to underestimate key tsunami characteristics such as runup and leading wave steepness. But the revised manuscript will also show how its simplified strategy serves immediate needs in greater Karachi.

- Tsunami preparedness in greater Karachi is overseen by an agency of Sindh Province, the Provincial Disaster Management Authority (PDMA). PDMA has thus far used deterministic assessment of tsunami hazards (Provincial Disaster Management Authority (PDMA) Sindh, 2024). The agency has considered three earthquake scenarios (Mw 9.0, 8.5, and 8.0). The 2024 plan includes community-level evacuation drills, early warning systems, and safe zones based on uniform slip models, offering clear and actionable insights for public awareness and preparedness. Our manuscript is intended to vet and underpin these efforts.

- The seven scenario earthquakes in the reviewed manuscript range from Mw 8.1 (emulating the observed Makran earthquake of 1945) to Mw 9.2 (as in the Omani maps). The Mw 9.2 scenario is a near-worst case taken directly from a widely cited geophysical report published a decade ago (Smith et al., 2013). In this Mw 9.2 scenario, the Makran subduction thrust ruptures across a down-dip extent of 350 km, from far inland to the deformation front. Rupture near the deformation front, where the water is deep, makes the tsunami large. Reviewer #3 notes

geodetic evidence against extending any eastern Makran scenario rupture far inland, but that onshore part of the rupture has no effect on tsunami generation.

- The reviewed manuscript, in its supplementary Table S1, provides an extensive review of previous reports on Makran tsunami hazards. Revised Table S1 will include additional reports to which Reviewer #3 points, and the text will reflect their content in recommending that future assessments of Karachi tsunami hazards incorporate splay faults, submarine landslides, and logic trees.

**INTENDED REVISIONS**

1. Clarify the concept of a tsunami mid-field as it applies to Karachi:
   We will refine the definition of "mid-field" for Karachi, clarifying its intermediate position between near-field and far-field tsunami zones. To support this, examples from the 2004 Indian Ocean tsunami (Sri Lanka, Thailand) will be added. Additionally, we will emphasize the limitations of Karachi's current warning system, highlighting challenges in issuing timely tsunami alerts for mid-field areas.

2. Note present-day uses elsewhere of deterministic modelling and of scenarios with uniform slip:
   We will address concerns about using "dated" methods by referencing current deterministic modelling practices from Chile, Cascadia, and Oman. These examples show that uniform slip scenarios are still widely used in official tsunami hazard assessments.
   The revised manuscript will update supplementary Table S1 to include additional reports cited by Reviewer #3. These updates will also reflect Reviewer #3's suggestions for future assessments of Karachi tsunami hazards to incorporate splay faults, submarine landslides, and logic trees, aligning with more recent advancements in the field.

3. Explain relevance of the work to emergency management in greater Karachi:
   We will revise the manuscript to clarify how our deterministic modelling approach directly addresses Karachi's immediate need for tsunami hazard maps. These maps will support the Provincial Disaster Management Authority's (PDMA) efforts in disaster preparedness and evacuation planning.

**REFERENCES CITED**

[The final list may be shorter than this one.]

Borrero, J. C., Lynett, P. J., and Kalligeris, N.: Tsunami currents in ports, Phil. Trans. R. Soc. A., 373, 20140372, https://doi.org/10.1098/rsta.2014.0372, 2015.

Carvajal, M. and Gubler, A.: The Effects on Tsunami Hazard Assessment in Chile of Assuming Earthquake Scenarios with Spatially Uniform Slip, Pure and Applied Geophysics, 173, 3693–3702, https://doi.org/10.1007/s00024-016-1332-x, 2016.

ESCAP: Tsunami Early Warning Systems in the countries of the North West Indian Ocean Region with focus on India, Islamic Republic of Iran, Pakistan, and Oman Synthesis Report, 2017.

Fritz, H. M. and Okal, E. A.: Socotra Island, Yemen: field survey of the 2004 Indian Ocean tsunami, Natural Hazards, 46, 107–117, https://doi.org/10.1007/s11069-007-9185-3, 2008.

Geist, E. L. and Dmowska, R.: Local Tsunamis and Distributed Slip at the Source, pure and applied geophysics, 154, 485–512, https://doi.org/10.1007/s000240050241, 1999.

INP: Tsunami early warning system installed at PMD's Pasni office, Daily Times, June, 2021.

Provincial Disaster Management Authority (PDMA) Sindh: Tsunami Management and Response Plan, 2024.

Smith, G. L., McNeill, L. C., Wang, K., He, J., and Henstock, T. J.: Thermal structure and megathrust seismogenic potential of the Makran subduction zone, Geophysical Research Letters, 40, 1528–1533, https://doi.org/10.1002/grl.50374, 2013.

UNDP: Strengthening Tsunami and Earthquake Preparedness in Coastal Areas of Pakistan: Quarterly Progress Report January – March 2021, United Nations Development Programme (UNDP), Pakistan, 2021.

Witze, A.: Tsunami alerts fail to bridge the 'last mile,' Nature, 516, 151–152, https://doi.org/10.1038/516151a, 2014.